# Mechanical glass transition revealed by the fracture toughness of metallic glasses

Jittisa Ketkaew[1], Wen Chen[1], Hui Wang[2], Amit Datye [1], Meng Fan[1], Gabriela Pereira[3], Udo D. Schwarz[1], Ze Liu[4], Rui Yamada[5], Wojciech Dmowski[2], Mark D. Shattuck[1,6], Corey S. O'Hern [1,7,8], Takeshi Egami[2,9,10], Eran Bouchbinder [11] & Jan Schroers[1]

The fracture toughness of glassy materials remains poorly understood. In large part, this is due to the disordered, intrinsically non-equilibrium nature of the glass structure, which challenges its theoretical description and experimental determination. We show that the notch fracture toughness of metallic glasses exhibits an abrupt toughening transition as a function of a well-controlled fictive temperature ($T_f$), which characterizes the average glass structure. The ordinary temperature, which has been previously associated with a ductile-to-brittle transition, is shown to play a secondary role. The observed transition is interpreted to result from a competition between the $T_f$-dependent plastic relaxation rate and an applied strain rate. Consequently, a similar toughening transition as a function of strain rate is predicted and demonstrated experimentally. The observed mechanical toughening transition bears strong similarities to the ordinary glass transition and explains the previously reported large scatter in fracture toughness data and ductile-to-brittle transitions.

[1] Department of Mechanical Engineering & Materials Science, Yale University, New Haven, CT 06511, USA. [2] Department of Materials Science and Engineering, University of Tennessee, Knoxville, TN 37996, USA. [3] Department of Mechanical Engineer, Universidade de Itaúna, Itaúna, Minas Gerais 35680-142, Brazil. [4] Department of Engineering Mechanics, School of Civil Engineering, Wuhan University, 430072 Wuhan, China. [5] Frontier Research Institute for Interdisciplinary Science (FRIS), Tohoku University, Sendai 980-8578, Japan. [6] Department of Physics and Benjamin Levich Institute, City College of the City University of New York, New York 10031, USA. [7] Department of Physics, Yale University, New Haven, CT 06511, USA. [8] Department of Applied Physics, Yale University, New Haven, CT 06520, USA. [9] Department of Physics and Astronomy, University of Tennessee, Knoxville, TN 37996, USA. [10] Oak Ridge National Laboratory, Oak Ridge, TN 37831, USA. [11] Chemical and Biological Physics Department, Weizmann Institute of Science, 7610001 Rehovot, Israel. Correspondence and requests for materials should be addressed to J.S. (email: jan.schroers@yale.edu)

The fracture toughness quantifies a material's ability to resist catastrophic failure in the presence of a crack. It is of enormous practical importance, as it is a major limiting factor in the structural integrity of a broad range of natural and engineering systems, and of great fundamental importance, as it challenges our understanding of the strongly non-linear and dissipative response of materials under the extreme conditions prevailing near defects[1]. Understanding the fracture toughness of glassy materials, which lack long-range crystalline order and are characterized by intrinsically disordered non-equilibrium structures, is a pressing problem in general and particularly for metallic glasses (MGs)[2].

MGs constitute a relatively new and broad class of amorphous materials with a combination of plastic-like processability and exceptional strength and elasticity—superior to their crystalline counterparts—holding great promise for wide-ranging structural and functional applications[3–6]. A major impediment, however, for their widespread usage as structural materials is not their strength, but rather their often low and highly variable fracture toughness[3].

Progress in understanding, predicting, and controlling the fracture toughness of MGs has been, on the whole, limited by the lack of: first, a theoretical understanding of the intrinsically disordered, non-equilibrium glassy states of matter; second, techniques to generate well-reproduced, well-defined glassy states; and third, accurate and reproducible fracture toughness samples and measurements. Glasses are non-equilibrium materials featuring disordered atomic structures whose properties are processing and history dependent[6,7]. From a theoretical perspective, currently there exists no general framework to quantify the disordered atomic structures of glasses and no complete understanding of the relation between these structures and glassy dynamics, most notably, irreversible plastic deformation that occurs in response to external driving forces. From an experimental perspective, carefully and reproducibly controlling the state of a glass and realizing these states in mechanical test samples has been very challenging, and has hampered accurate and reproducible fracture toughness measurements[2].

Here, we show that the notch fracture toughness of MGs exhibits an abrupt toughening transition as a function of the fictive temperature ($T_f$), whereas the ordinary temperature plays a secondary role. This observed mechanical toughening transition, which we theoritically explain, bears strong similarities to the ordinary glass transition and explains the previously reported large scatter in fracture toughness data and ductile-to-brittle transitions. The presented results open the way for a broader usage of tougher and well-reproducible MGs as structural materials.

## Results

### Fracture toughness sample synthesis of well-controlled $T_f$.
We use a thermoplastic forming (TPF) method that allows precise and reproducible measurements of the notch fracture toughness. Specifically, we thermoplastically mold MGs into single edge notch tension (SENT) fracture toughness samples. The selected geometry and particular notch radius of the samples were kept constant for all of the experiments reported in this work (see Methods). The glass was then heated to a temperature $T_f$, in the vicinity of the calorimetric glass transition temperature ($T_g$), and annealed at this temperature for a time that exceeds the structural relaxation time ($\tau_{SR}$) at $T_f$, ensuring full equilibration (see Methods). Subsequently, the glass was rapidly quenched to room temperature at a rate exceeding $(T_f - T_g)/\tau_{SR}$ in order to minimize structural relaxation during the quench. The fictive temperature is also calculated through heat capacity

measurements to verify that it identifies with the annealing temperature for the considered protocol[8,9] (see Supplementary Fig. 6).

This protocol yields glasses whose metastable structural state is well characterized by $T_f$—a temperature characterizing the structural degrees of freedom of the glass where it has fallen out-of-equilibrium (often referred to as the fictive temperature[10,11] or glass-transition-upon-cooling)—which is different from the calorimetric glass transition temperature, $T_g$. $T_f$ is set by the annealing time and is maintained through fast cooling to avoid relaxation to a lower $T_f$ upon cooling to the test temperature $T$. $T_g$, as typically used for MGs, is determined upon heating with rates of typically 20 K/min (0.3 K/s). The magnitude of thermal fluctuations, on the other hand, is determined by the ordinary temperature ($T$). Whereas the ordinary temperature $T$ characterizes the vibrational degrees of freedom of the glass that quickly equilibrate with the heat reservoir, $T_f$ encapsulates the glass structure which is affected by its annealing and fabrication history. Our preparation protocol results in highly precise and reproducible glassy states, and consequently toughness measurements, with insignificant sample-to-sample variations. Measurements of the notch fracture toughness ($K_Q$) were carried out for glasses with $T_f$ varied by ~100 K around $T_g$, which corresponds to more than six orders of magnitude in relaxation times, and $T$ was varied over ~500 K, from 77 K to 573 K, and at a given applied strain rate ($\dot{\varepsilon}$) as a function of $T_f$. These measurements allow us to disentangle the dependence of $K_Q$ on a well-defined and controlled non-equilibrium structural state of a glass encoded in $T_f$ and on thermal vibrations quantified by $T$, probing the unique properties of glasses. Three MGs—$Zr_{44}Ti_{11}Ni_{10}Cu_{10}Be_{25}$ ($T_g = 623$ K), $Pd_{43}Cu_{27}Ni_{10}P_{20}$ ($T_g = 578$ K), and $Pt_{57.5}Cu_{14.7}Ni_{5.3}P_{22.5}$ ($T_g = 503$ K)—were studied. These MGs were selected since they include representatives of the two primary classes of MGs, metal–metal and metal–metalloid alloys[6], which exhibit a wide range of toughness and fragility levels[12], and varying degrees of $\beta$ relaxation[13].

### Abrupt transition of the fracture toughness as a function of $T_f$.
We first determined the fracture toughness as a function of the fictive temperature by fixing $T = 300$ K (room temperature), well below $T_g$ for all three MGs, and varied $T_f$. $K_Q(T = 300$ K, $T_f)$ for a fixed strain rate of $10^{-4}$/s for all three glasses is shown in Fig. 1a. $K_Q$ exhibits an abrupt toughening transition between a brittle-like regime below a threshold value of $T_f$ ($T_f^{DB}$) and a ductile-like regime above it. The degree of toughening across the transition is dramatic for all three MGs and can be as high as ~260% (for $Zr_{44}Ti_{11}Ni_{10}Cu_{10}Be_{25}$). Such transition as a function of the fictive temperature, which is our main finding, has not been previously reported and is significantly more pronounced than the ductile-to-brittle transition observed for different ordinary temperatures (Fig. 1b)[14–16]. Note, though, that earlier work did indicate that structural relaxation affects the ductility of MGs[17]. The MG-specific threshold for which $K_Q(T_f)$ starts to increase rapidly is surprisingly close to $T_g$ for all three MGs, which may already suggest some relation to the ordinary glass transition. The transition is accompanied by a significant increase in the plastic zone size prior to catastrophic failure, as shown in Fig. 1c for $Zr_{44}Ti_{11}Ni_{10}Cu_{10}Be_{25}$. We hypothesize that the brittle-like and ductile-like behaviors observed in Fig. 1a are associated with different microscopic failure mechanisms, which should have distinct fractographic signatures below vs. above the transition. To test this hypothesis, we performed scanning electron microscope scans of the post-mortem fracture surfaces. The resulting images for $Zr_{44}Ti_{11}Ni_{10}Cu_{10}Be_{25}$ are shown in Fig. 1d for the same $T_f$ values considered in Fig. 1c. The revealed fracture surface morphology exhibits a marked transition from dimple structures,

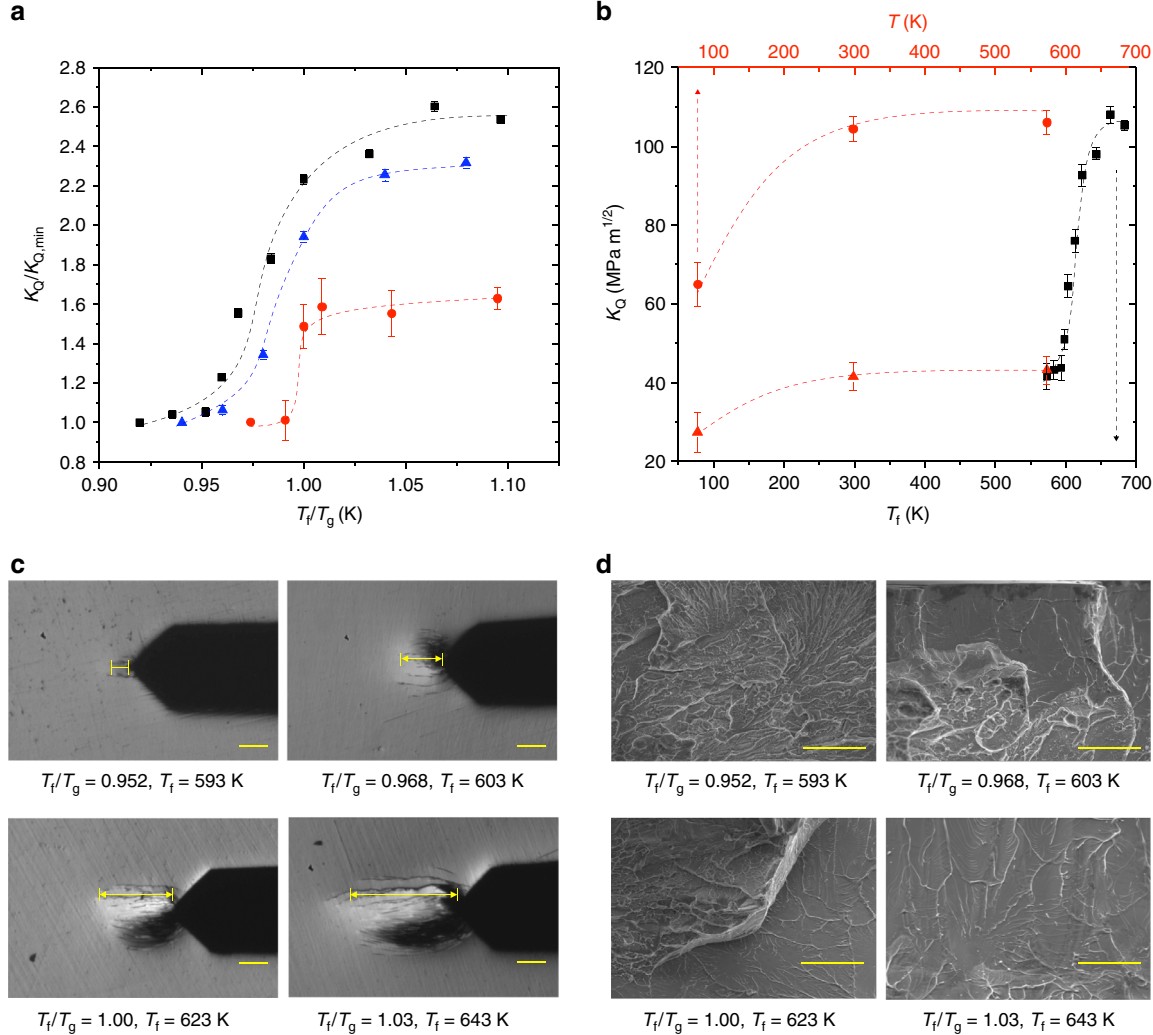

**Fig. 1** The notch fracture toughness of metallic glasses exhibits a dramatic transition as a function of fictive temperature, $T_f$. **a** The notch fracture toughness $K_Q$, normalized by its minimal value $K_{Q,min}$, as a function of $T_f$, normalized by the glass temperature $T_g$, for $Zr_{44}Ti_{11}Ni_{10}Cu_{10}Be_{25}$ (black squares), $Pd_{43}Cu_{27}Ni_{10}P_{20}$ (red circles), and $Pt_{57.5}Cu_{14.7}Ni_{5.3}P_{22.5}$ (blue triangles). The error bars represent 1 standard deviation of five samples per data point. The dashed lines serve as a guide to the eye. **b** $K_Q$ for $Zr_{44}Ti_{11}Ni_{10}Cu_{10}Be_{25}$ as a function of $T_f$ (measured at room temperature, black symbols—bottom axis) and $T$ (red symbols—top axis, measured at $T$) with $T_f = 683$ K $> T_f^{DB}$ (red circles) and $T_f = 583$ K $< T_f^{DB}$ (red triangles). The dashed lines represent polynomial fits of the data. **c** The plastic zone ahead of the notch root just prior to failure of $Zr_{44}Ti_{11}Ni_{10}Cu_{10}Be_{25}$ for several $T_f$. Dimension lines indicate the plastic zone size. For $T_f < T_f^{DB}$, the plastic zone is small, while for $T_f > T_f^{DB}$ it is significantly larger. The plastic zone is symmetric with respect to the main axis of the notch (darker areas are optical effects). The scale bars are 100 μm. **d** The fracture morphology of $Zr_{44}Ti_{11}Ni_{10}Cu_{10}Be_{25}$ for various $T_f$. As the threshold $T_f^{DB}$ is surpassed (top right), the fracture morphology changes from fractal-like structures (characteristic of brittle-like fracture) to river-like patterns (characteristic of ductile-like fracture). The scale bars are 50 μm

previously reported to be associated with brittle fracture[18], below $T_f^{DB}$, to river-like patterns above $T_f^{DB}$, which have been associated with ductile fracture[19]. This change in fracture surface morphology coincides with the transition in $K_Q$ observed in Fig. 1a. In fact, quite remarkably, at the transition, dimple structures and river-like patterns appear to coexist (sub-panel for $T_f/T_g = 1$ in Fig. 1d). A similar fractographic behavior of the fractured surface of $Pd_{43}Cu_{27}Ni_{10}P_{20}$ and $Pt_{57.5}Cu_{14.7}Ni_{5.3}P_{22.5}$ was also observed (Supplementary Fig. 1).

**Comparison of the $T_f$ and $T$ dependence of the fracture toughness.** One may ask whether the observed toughening (ductile-to-brittle) transition that occurs as a function of $T_f$ (Fig. 1a) is related to the previously reported ductile-to-brittle transition in MGs that occurs as a function of the ordinary temperature $T$[13,17,20]. We directly compare the dependence of $K_Q$ on $T$ in Fig. 1b to the dependence of $K_Q$ on $T_f$ in Fig. 1a by also

measuring $K_Q$ as a function of $T$ for $Zr_{44}Ti_{11}Ni_{10}Cu_{10}Be_{25}$, spanning a temperature region from 77 to 573 K. Similar experiments were also conducted for $Pd_{43}Cu_{27}Ni_{10}P_{20}$ (Supplementary Fig. 2). We find that the variation of $K_Q$ with $T_f$ is significantly larger than the negligible variation of $K_Q$ with $T$ over the same temperature range (Fig. 1b), highlighting the structural nature of the transition. These results lead us to conclude that the fracture toughness of MGs is qualitatively and dramatically more sensitive to the non-equilibrium structural state of the glass quantified by $T_f$ than to the (ordinary) temperature $T$ (at least down to very low ordinary temperatures compared to $T_f^{DB}$, where another transition might take place, see the red data points at the liquid nitrogen temperature in Fig. 1b).

**Variation of structural and response quantities with $T_f$.** To understand the origin of the abrupt and dramatic transition as a function of $T_f$, we studied other structural and response quantities

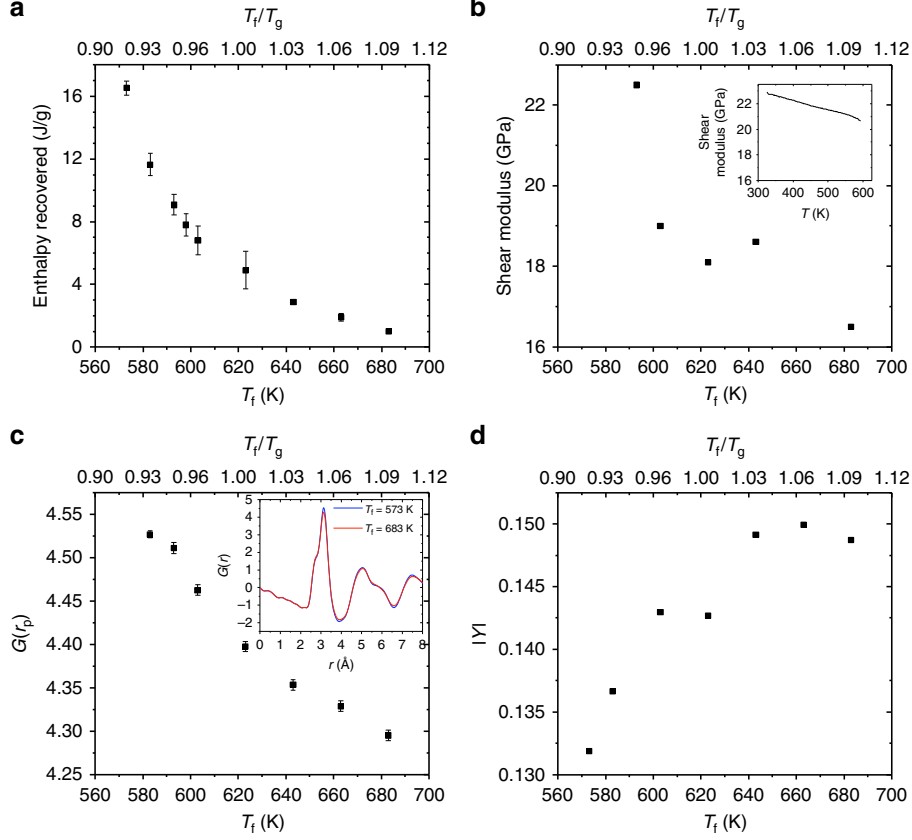

**Fig. 2** Structural and response quantities of the $Zr_{44}Ti_{11}Ni_{10}Cu_{10}Be_{25}$ metallic glass exhibit gradual changes as a function of $T_f$. **a** Enthalpy recovery as a function of $T_f$ obtained from differential scanning calorimetry of SENT fracture toughness samples. The error bars are quantified by 1 standard deviation from three samples. **b** The shear modulus $G$ as a function of $T_f$, obtained from dynamic mechanical analysis at room temperature. (inset) $G$ as a function of $T$ at fixed $T_f = 593$ K. **c** The amplitude of the first peak of the pair distribution function $G(r_p)$ as a function of $T_f$. (inset) The pair distribution function $G(r)$ for the two extreme cases: $T_f = 573$ and 683 K. **d** The anisotropy $|Y|$ of the pair distribution function under a compressive stress of 1 GPa as a function of $T_f$ (see Methods for the precise definition of $|Y|$)

of glasses as a function of $T_f$. Specifically, we quantified the enthalpy recovery, the shear modulus ($G$), and the atomic pair distribution function ($G(r)$), and its response to applied stress[21,22] for $Zr_{44}Ti_{11}Ni_{10}Cu_{10}Be_{25}$ as a function of $T_f$. Enthalpy recovery, which has been associated with free volume[23] and the ductile-to-brittle transition[17], varies significantly, by more than an order of magnitude over the $T_f$ range we considered (Fig. 2a). However, this variation occurs smoothly, without abrupt changes, particularly near $T_f/T_g = 0.98$, where $K_Q$ exhibits a strong variation with $T_f$ (cf. Fig. 1a). $G$, which has been widely associated with ductility and fracture toughness of MGs[24], decreases gradually as $T_f$ increases (Fig. 2b). The inset shows the variation of $G$ with $T$ at a fixed $T_f = 593$ K, which is one order of magnitude smaller than the reduction in $G$ as a function of $T_f$. High-energy X-ray diffraction measurements were used to extract the radial distribution function $G(r)$ as a function of $T_f$ (inset of Fig. 2c). More information on the structure function and pair distribution function of different fictive temperatures glasses is shown in Supplementary Figs. 3–4. The amplitude of the first peak of $G(r)$, shown in Fig. 2c, exhibits a mild and gradual variation with $T_f$. The elastic heterogeneity of $Zr_{44}Ti_{11}Ni_{10}Cu_{10}Be_{25}$ under uniaxial stress was measured through the anisotropy of the pair distribution function ($|Y|$)[25] (see Methods for a precise definition of $|Y|$). $|Y|$, shown in Fig. 2d, also exhibits a rather mild and gradual variation with $T_f$. Therefore, we conclude that these structural and response quantities do not reveal any signature of the abrupt increase as a function of $T_f$ that is exhibited by $K_Q$.

**Mechanical glass transition**. What is then the physical origin of the $T_f$-dependent toughening transition observed in Fig. 1a? On the one hand, the transition is clearly sensitive to the initial non-equilibrium structural state of the glass, as quantified by $T_f$ (Fig. 1a). On the other hand, several structural and response quantities (Fig. 2), which are often associated with the toughness of glasses, do not change significantly around a MG-specific value of $T_f^{DB}$. Taken together, these results suggest that the initial non-equilibrium structural state of the glass plays a crucial, but not exclusive, role in the observed toughening transition. Following the theoretical work of refs. [26,27], we propose that the origin of this toughening transition is a competition between two time scales. The time scales involved are the plastic deformation time scale ($\tau_{plastic}$), which controls the plastic dissipation in the vicinity of the notch where stresses are close to the yield strength, and the loading time scale ($\tau_\dot{\varepsilon}$), which is inversely proportional to the applied strain rate. $\tau_{plastic}$ is inversely proportional to the density of plasticity carriers (i.e., shear transformation zones, sometimes also related to concepts such as soft spots[28,29], flexibility volume[30], core-shells[31], and flow units[32–34]), which is a strongly increasing function of $T_f$ and a much weaker increasing function of $T$[27]. It has been proposed that $\tau_{plastic}$ also depends on the local stress, as it provides the activation energy for plastic rearrangements[27,35].

Indeed, we found that glasses which are tested at significantly different temperatures $T$ fail under the same macroscopic conditions, indicating that the aforementioned dependence of

$\tau_{plastic}$ on the local stresses and $T_f$ is much stronger than on $T$. Specifically for $Zr_{44}Ti_{11}Cu_{10}Ni_{10}Be_{25}$ with $T_f = 593$ K and $T = 593$ K, we measured $K_Q = 45.2 \pm 3$ MPa m$^{1/2}$, which is, within experimental error, identical to $K_Q = 43.2 \pm 3$ MPa m$^{1/2}$ measured for $Zr_{44}Ti_{11}Cu_{10}Ni_{10}Be_{25}$ at $T_f = 593$ K and $T = 300$ K (see empty circle symbols in Fig. 1b). As the local stresses (set by the sample's geometry and the external loading) and $T$ are identical in both cases these results reveal a significantly higher sensitivity of $\tau_{plastic}$, and hence of $K_Q$, to the intense stresses near the notch and $T_f$ compared to $T$, which has been also observed for $Pd_{43}Cu_{27}Ni_{10}P_{20}$ MG.

The time scale that competes with $\tau_{plastic}$ is the inverse of the strain rate, $\dot{\varepsilon}$, in the plastic zone, $\tau_{\dot{\varepsilon}}, \tau_{\dot{\varepsilon}} \propto \dot{\varepsilon}^{-1}$. In the brittle-like regime, we expect the glass response to be predominantly elastic, with limited plastic relaxation of stresses, corresponding to $\tau_{plastic} \gg \tau_{\dot{\varepsilon}}$. In the ductile-like regime, we expect more extensive plastic deformation, corresponding to $\tau_{plastic} \ll \tau_{\dot{\varepsilon}}$. As $\tau_{\dot{\varepsilon}}$ is independent of $T_f$ and $\tau_{plastic}$ is a strongly decreasing function of $T_f$, an abrupt toughening transition, qualitatively similar to the one observed experimentally (Fig. 1a), is expected to occur when $\tau_{plastic} \approx \tau_{\dot{\varepsilon}}$. Furthermore, as $\tau_{plastic}$ is also a function of $T$, yet a much weaker function compared to $T_f$[27], we expect a rather mild decrease of the toughness with $T$, consistent with previously reported ductile-to-brittle transitions, as well as with our $K_Q(T)$ data (Fig. 1b and Supplementary Fig. 2).

This theoretical picture suggests that the fracture toughness of glasses is not an entirely intrinsic material property, as it predicts a dependence on the externally applied strain rate $\dot{\varepsilon}$. The proposed crossover of time scales suggests that for fixed $T_f$ (and $T$), $K_Q$ exhibits an abrupt toughening transition when $\dot{\varepsilon}$ drops below a MG-specific threshold value. To test this important prediction, we performed measurements of $K_Q$ for the $Zr_{44}Ti_{11}$-$Ni_{10}Cu_{10}Be_{25}$ MG over a wide range of strain rates $\dot{\varepsilon}$ for two values of $T_f$ (both below and above the transition in Fig. 1a). The results, presented in Fig. 3, reveal a $T_f$-dependent toughening transition with decreasing $\dot{\varepsilon}$, as suggested by the proposed crossover of time scales.

These results suggest that the toughening transition observed in Figs. 1a and 3 may be viewed as a mechanical glass transition in analogy with the conventional glass transition (Fig. 4). With this interpretation, the role of the structural relaxation time, $\tau_{SR}$, in the conventional glass transition is played by $\tau_{plastic}$ and the role of the cooling rate, $\dot{R}$, in the conventional glass transition is played by $\dot{\varepsilon}$. The corresponding role of the thermodynamic quantity that is used to probe the conventional glass transition as a function of $T$, e.g., the enthalpy $H$, is $K_Q(T_f)$ in the mechanical glass transition.

## Discussion

Our results, in addition to their fundamental importance for glass physics, have significant practical implications as they offer a well-defined procedure to realize the practically maximal fracture toughness of MGs defined by their composition and by the strain rate in a specific application. Such realization can be achieved by carefully controlling $T_f$ through the annealing protocols described above. As a consequence, the observed $K_Q(T_f)$ defines a critical cooling rate (setting $\tau_{plastic}$) to achieve ductile behavior, distinct from the critical cooling rate for glass formation. Previously, it has been proposed to control the toughness of glasses by the cooling rate through the glass transition[36]. This procedure and other relaxation procedures, however, reported a gradual variations of the toughness or other properties like hardness[37], similar to those observed here for structure and response quantities (Fig. 2), in sharp contrast to dramatic toughening transition reported here (Fig. 1). Moreover, the abrupt toughening transition and the existence of $T_f^{DB}$ might be at the heart of the large scatter in the reported fracture toughness values for chemically identical MGs[2], which most likely were measured for different—

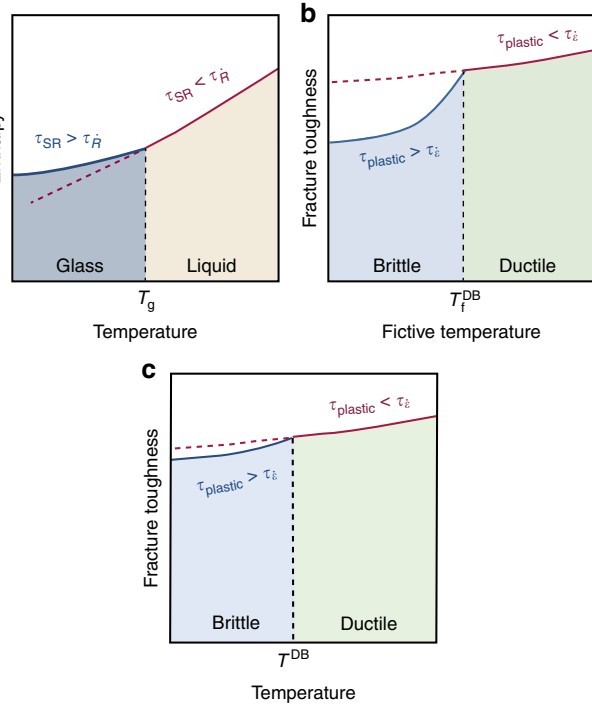

**Fig. 4** Analogy between the conventional glass transition and the mechanical glass transition based on a crossover of time scales. **a** The conventional glass transition, probed by the dependence of the enthalpy on $T$, originates from a competition between the internal structural relaxation time, $\tau_{SR}$, and the external time scale set by the cooling rate, $\tau_{\dot{R}}$. The glass transition occurs approximately at $\tau_{SR} \approx \tau_{\dot{R}}$. **b** In analogy to the conventional glass transition, the mechanical glass transition (probed by the dependence of the fracture toughness on $T_f$) originates from the competition between the plastic relaxation time scale, $\tau_{plastic}$, and the (near notch) deformation time scale, $\tau_{\dot{\varepsilon}}$, which is proportional to the applied strain rate. **c** A ductile-to-brittle transition can also be observed as a function of temperature, as $\tau_{plastic}$ is also a weak function of $T$, but it is significantly less pronounced than the toughening transition as a function of $T_f$

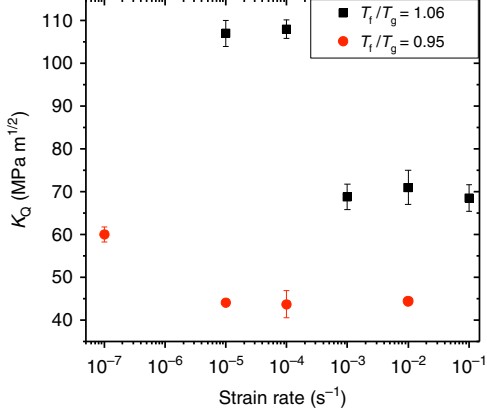

**Fig. 3** The fracture toughness of metallic glasses also exhibits a toughening transition with decreasing strain rate. $K_Q$ as a function of strain rate $\dot{\varepsilon}$ for $Zr_{44}Ti_{11}Ni_{10}Cu_{10}Be_{25}$ with $T_f/T_g = 1.06$ ($T_f > T_f^{DB}$, black squares) and $T_f/T_g = 0.95$ ($T_f < T_f^{DB}$, red circles). The error bars represent 1 standard deviation calculated from three samples

and uncontrolled—values of $T_f$. Since $T_f^{DB}$ is also strain rate dependent, and shifts to higher $T_f$ with increasing strain rate, MGs may behave significantly different in high and low strain rate applications. Taken together, our findings reveal and explain a mechanical glass transition, which should be integrated into glass theories and be technologically considered for a much broader usage of MGs as tough, highly reproducible structural materials.

## Methods

**Notch fracture toughness sample preparation.** Three bulk MG formers were utilized to study the effect of chemistry and fictive temperature on the notch fracture toughness, $Zr_{44}Ti_{11}Ni_{10}Cu_{10}Be_{25}$, $Pd_{43}Cu_{27}Ni_{10}P_{20}$, and $Pt_{57.5}Cu_{14.7}Ni_{5.3}P_{22.5}$. Amorphous $Zr_{44}Ti_{11}Ni_{10}Cu_{10}Be_{25}$ was obtained from Materion. $Pd_{43}Cu_{27}Ni_{10}P_{20}$ and $Pt_{57.5}Cu_{14.7}Ni_{5.3}P_{22.5}$ were prepared by induction melting the constituents in quartz tubes and subsequently fluxed in $B_2O_3$ at 1350 K. The flux materials were then removed and the alloys were re-melted and cast into the amorphous state by rapid water quenching. MGs were formed into SENT samples, which were fabricated by a TPF process into silicon molds. SENT samples were designed in Layout Editor software and were transferred onto the photomask. A layer of positive photoresist was spun on a silicon wafer of thickness ~10 μm. The wafer was etched as defined by the design using a deep reactive ion etching process to a final depth of 350 μm. TPF of SENT samples was accomplished by heating the silicon mold and MG into the supercooled liquid region under applied pressure. Excess material was removed by sanding. All samples were polished to a 1 μm finish after annealing to maintain the same roughness prior to mechanical testing.

**Fictive temperature manipulation.** Liquids from above $T_m$ are cooled to glassy states at $T_{f,1}$. To ensure same thermal history of the glass phase, all samples were heated to $T_g + 10$ K. SENT samples were brought to a given fictive temperatures by annealing the glass at various temperatures with 1.5 times the relaxation time defined by the Vogel–Fulcher–Tammann (VFT) relation to ensure the new equilibrium at $T_f$ has been reached[38]. The VFT parameters can be obtained by differential scanning calorimetry (DSC) experiments in combination with fitting the VFT equation $\tau = \tau_0 \exp(\frac{D^* T_0}{T - T_0})$, where $\tau$ is the relaxation time at $T$ and $\tau_0$ is the infinite temperature relaxation time. DSC experiments were employed to determine $D^*$ and $T_0$ as described by Launey and Kruzic et al[39–41]. Detailed of the thermal history of samples is shown in Supplementary Fig. 5. Determination of fictive temperature through specific heat measurements[8,9] is also explained in Supplementary Method.

**Fracture toughness measurements.** Mechanical testing of as-fabricated SENT samples with different $T_f$ was tested using uniaxial tensile testing with quasi-static displacement control with an initial strain rate of $10^{-4}\,s^{-1}$. Here, we measure conditional (notch) fracture toughness, $K_Q$, and not $K_c$ for two reasons. First, since MG's metastable nature limits their casting dimension, it is generally difficult to obey the standard $K_{IC}$ procedure. Second, the objective of this study is to understand the dependence of $T_f$ on the relative change of toughness, which required decoupling this effect from other scattering effects. However, absolute values of fracture toughness reported here may have similar limitations as previous measurements of the fracture toughness of MGs. $K_Q$ was obtained using the relation $K_Q = \sigma\sqrt{\pi a}F(\frac{a}{W})$, where $\sigma$ is the applied far-field stress, $a$ is the notch length of 2.5 mm, notch root radius of 10 μm, $w$ is the sample width of 5 mm, and the geometry factor $F(\frac{a}{W}) = \sqrt{\frac{2W}{\pi a}\tan\frac{\pi a}{2W}} \cdot \frac{0.752 + 2.02(\frac{a}{W}) + 0.37(1 - \sin\frac{\pi a}{2W})^3}{\cos\frac{\pi a}{2W}}$. According to the guidelines provided by ASTM E399 for a plain-strain $K_{IC}$, the thickness $t$, the notch length $a$, and the length of the uncracked ligament $(W - a)$ need to satisfy the following condition, $a, t, W - a \geq 2.5\left(\frac{K_c}{\sigma_y}\right)^2$, where $\sigma_y$ is the yield strength of the MG. For example, the MG $Zr_{44}Ti_{11}Ni_{10}Cu_{10}Be_{25}$ has yield strength $\sigma_y = 1.9$ GPa. Hence, the suggested geometric requirement is $a, t, W - a \geq 8$ mm, which is difficult to realize for most MGs. Therefore, we measure $K_Q$ instead. In addition, we have shown in ref. [42] that $K_{IC}$ can be extracted from our $K_Q$ measurements.

**Atomic structure, elastic constant, and thermal characterization.** As-cast material and all test samples were confirmed to be in an amorphous state by Rigaku Smartlab X-ray diffraction. Standard thermal analysis and enthalpy recovery experiments were performed by Perkin Elmer diamond DSC with a heating rate of 20 K/min. The heat capacity ($C_p$) was measured as a function of temperature to estimate the $T_f$ values. The samples were heated from 313 K after equilibration for 300 s to 663 K at a heating rate of 10 K/min, then let equilibrated for 2 min. Samples were then cooled to 573 K with a cooling rate of 10 K/min and then cooled to 313 K at 20 K/min. Samples were subjected to a second run with the repeating procedure. Based line measurements were carried out, and a sapphire sample was used as a reference for the $C_p$ measurement. All the measurements were done under an argon atmosphere.

Shear moduli were obtained from dynamic mechanical testing using a TA Instruments Ares G2 performed using the torsion clamps. The gauge lengths of the samples varied from 8 to 12 mm. The samples were tested at 0.01% strain at 1 Hz frequency from 320 to 770 K at a heating rate of 5 K per min. An auto tension of 0.1 N (≈10 g) was used to ensure proper measurements.

High-energy X-ray diffraction for characterization of atomic structure was carried out at the Advanced Photon Source (Argonne National Laboratory) beamline 1-ID and 6-ID. The incident energy was tuned to 100 keV and a 2D area detector was placed ~34 cm behind the sample. Calibration was performed using the $CeO_2$ NIST powder standard. High-energy X-ray diffraction data were processed by FIT2D software[43]. The MTS loading frame was used for the in situ structural study under uniaxial compression (see Supplementary Method for details on in situ structural study).

**Data availability**. All data generated or analyzed during this study are included in this published article (and its Supplementary Information).

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

## Acknowledgements

We warmly thank Prof. Frans Spaepen for fruitful discussions. This work was supported by the U.S. Department of Energy through the Office of Science, Basic Energy Sciences, Materials Science and Engineering Division (No. DE SC0004889). Structural characterization was carried out at the Advanced Photon Source, a U.S. Department of Energy (DOE) Office of Science User Facility, operated for the DOE Office of Science by Argonne National Laboratory under Contract DE-AC02-06CH11357. E.B. acknowledges support from the Richard F. Goodman Yale/Weizmann Exchange Program. A.D. acknowledges support by the Department of Energy through grant No. DE-SC0016179.

## Author contributions

J.S., T.E., E.B., J.K., Z.L., and W.C. designed and developed the study. J.K., R.Y., G.P. conducted the main experiments (sample preparation, TPF, fracture toughness, and DSC). H.W. and W.D. conducted the X-ray diffraction experiment and analyzed the results. A.D. and U.D.S. conducted DMA experiment and analyzed the data. E.B., M.F., M.D.S., and C.S.O. developed the theoretical model. J.K., J.S., E.B., T.E. analyzed the data and wrote the manuscript. All authors contributed to the discussion of the results and revised the manuscript.

## Additional information

**Competing interests:** The authors declare no competing interests.

