## [Peer Review File · Nature Communications]

Reviewers' comments:

Reviewer #1 (Remarks to the Author):

The authors studied the fictive temperature dependence of fracture toughness of some metallic glasses and linked it to the structural changes. They measured conditional (notch) fracture toughness, KQ for different T_f , and found a characteristic fictive temperature where fracture toughness exhibits non-trivial and abrupt increase from brittle to ductile behavior. They attribute this change in fracture toughness to the competition between two different time scales. They regarded this transition as a mechanical glass transition. I should mention that such transition has been reported in literature also for other glass formers. The results are interesting and has practical implications. The paper is well written. But the discussion is not convincing in several places. I list some specific problems below:

1. In Discussion, the author wrote "we argue that the rather abrupt change in $KQ(T_f)$ emerges from a competition of time scales." Although Fig. 4a shows the different KQ for three different strain rates, the competition of τ and τ_{plastic} is not shown. In Ref [59], there are some specific explanations about notch toughness as a function of ξ . But this work involves some speculations and lacks clear evidences.
 2. Detailed description about Fig. 4b is needed to make the figure more understandable.
 3. As shown in Fig. 4c-g, the authors performed MD simulations. But the simulation details in Methods part are not clear enough. Besides, there are no explanations about the information in these figures. The meaning of parameter $P(\Delta r)$ are not mentioned, though it can be interpreted according to last part of Methods.
 4. T_f was regarded as the annealing temperature at which the samples were relaxed with the duration of 1.5τ . The physical meaning of their T_f has not been described. What is the difference between their T_f and enthalpic T_f determined by DSC? It would be helpful if they can compare both T_f s. The DSC methods for determining enthalpic T_f can be found in literature.
 5. The authors should provide the references for the dependence of relaxation time on temperature in the Pd-based and Pt-based MGs?
 6. The T_g values for all samples should be given in the manuscript.
 7. The authors should use Kelvin as the unit of the T_f and T_g should be. Only using Kelvin, would T_f/T_g have physical meaning.
 8. Only 4 points in Figure 2 were plotted in the Pt-based MG, which is not convincing to support the abrupt changes of the fracture toughness around T_g .
 9. Line 38: molecular liquids should be molecular glasses.
- In summary, this paper is not suitable for Nature Communication regarding its originality and general interest. It might suit a materials science journal.

Reviewer #2 (Remarks to the Author):

The great strength of this work is the quality of the data. As shown in earlier work by some of the authors, they have an experimental method by which the notch fracture toughness (KQ) can be measured with a variability of less than 3%. This is remarkable and useful. The authors have also developed a method by which metallic glass samples can be prepared with controlled values of fictive temperature (T_f).

The main contribution that the authors claim for this work is that the transition in KQ (from a low value to a high value as T_f is increased) is sharp, not gradual. It is good that the authors have been able to better characterize this transition and to show that it is sharp – but this is certainly not a new or unexpected result.

The authors are entirely correct (line 145) to draw a distinction between the change in KQ at a critical value of T_f and the classical ductile/brittle transition as a function of temperature. Yet the

two behaviors are linked. They both show a transition from brittle fracture to ductile failure as a parameter is changed. That parameter can be internal to the glass (T_f), or external (test temperature or strain rate). The authors are correct to focus on the importance of relative time scales (lines 206-223). But for a given applied strain rate, the transition in failure mode can be considered in terms of stress. At a given strain rate (and with other parameters such as T_f and test temperature fixed) there will be a critical stress for the onset of brittle fracture and a (different) critical stress for the onset of ductile failure. The question, rather simply, is which stress is lower? The lower stress, of course, corresponds to the failure mode that will be observed. As some parameter (for example, test temperature or T_f) is changed, the critical stress values will change and which stress is lower may change. If the identity of the lower stress does change, the failure mode changes, and inevitably there is a rather sharp change in toughness. The point is that transitions in toughness are expected to be sharp, associated with a change in failure mode. Could it be that earlier work suggesting smoother transitions suffered from scatter in the measurements of toughness?

In lines 173 and 174, the authors note that “none of these quantities exhibit a T_f dependence similar to that of KQ ”. This point is reinforced in lines 202 and 203 – but there is nothing remarkable about this. The various quantities (G/B , enthalpy recovery, and static atomic pair correlation function and its response to stress) all show (according to the authors’ own data, Fig. 3) a smooth correlation with T_f . Thus any of these quantities could be used as a proxy for T_f (or T_f/T_g) in plotting a figure such as Fig. 2a. And, to repeat the point made in the previous paragraph, it is straightforward to understand why the toughness shows a sharp transition as a function of any of these variables.

In the paragraph starting on line 257, the authors consider “enthalpic contributions” to KQ . This approach is at risk of confusing thermodynamic and kinetic aspects. As the authors themselves note, there are potential contributions from thermal activation, so that kinetics could be important. The particular example cited by the authors is interesting, but it would be misleading to assume that it is general. Specifically, the authors find that KQ for a glass with $T_f = 300$ degC and measured at 300 degC is the same as for a glass with $T_f = 300$ deg C measured at room temperature. This does not, however, say that the test temperature is unimportant (which, I think, is what the authors are implying when they discount the influence of H_{volume}). (This is also implied in the comment about “Enthalpic changes” in lines 306 and 307.)

A glass with a given structure (and given T_f) will have a ductile/brittle transition temperature. Well above or below that temperature, KQ is likely to be rather constant, independent of temperature. Thus the similar values of KQ noted in the previous paragraph simply tell us that room temperature and 300 degC are on the same side of the transition temperature; they do not imply that, overall, “contributions from volume expansion and thermal excitation are negligible” (line 265).

On line 302, the authors state that a transition from brittle-like to ductile-like behavior “is absent in conventional models”. This statement seems too strong, given the overall interest in transitions in the failure mode.

Minor points:

In line 134, the authors discuss “ KQ at T_g ”. That would normally mean ‘ KQ measured at T_g ’, which is not what the authors intend. Instead, they mean something like: “ KQ at $T_f=T_g$ ”.

In Fig. 2b, the images may be too small to be useful.

Line 331: should be ‘Tammann’

Overall -- really good data giving a comprehensive characterization of the ductile/brittle transition, but not offering new physical insights.

Reviewer #3 (Remarks to the Author):

The paper proposed by Ketkaew et al. on the control of ductility in MG by fictive temperature is, in my opinion, definitely worthy of publication in Nature. Indeed, it establishes for the first time a rather revolutionary and fundamental fact, based on solid experimental - and original - results along with interesting discussion: the fictive temperature control the glass toughness. It opens new and very wide opportunities for academic research in all of glass science (a priori not limited to metallic glasses) and will have important implications for the use of BMG as structural materials, in particular concerning the heat treatments that should be applied to improve their toughness.

The main finding of the paper is that MG ductility is strongly controlled by their fictive temperature, in a very systematic way, on three different compositions (Pt, Pd and Zr based MG) with a particular care taken about the meaningfulness of the results (which are definitely meaningful: no experimental error seems able to account in any way for the results which are extremely consistent). This finding has never been reported elsewhere, to the best of my knowledge, and this very question is addressed for the first time, as the experimental control is tricky and needs the great experience the authors have. What's more, the discussion is based on additional structural analysis that opens fundamental implications of prime importance in the field (notably the variation with temperature of G/B, recovery enthalpy, $G(rp)$ and $|Y|$ and their correlation with Kq).

The main claims are definitely convincing, but some aspect could, in my opinion, be improved or nuanced without great effort, I'd say, as it would not involve any additional experimental work. So, however superb the results and the work, I do have a few points which, for some of them, seem significant, and also a few questions. I present my points as the text goes, not by order of importance:

- 1) In line 85 and 86, the authors state that they "provide experimental evidence that upon cooling from T_f to room temperature [...] the glass remains in a single 'meta basin' in the energy landscape." : I must say that this point is actually not made clearly enough in the text (or I may have missed it) and it would be worth being made clearer because of its importance.
- 2) Line 136-139, the authors mention that for ZrTiCuNiBe, the fracture pattern evolves significantly and they show said evolution in figure 2.c. It would be worth at least discussing what they have observed for the other two glasses, and possibly include SEM images (if available) in the supplementary material. On the side note, figure 2.c should be made larger for comfort of viewing.
- 3) The authors stress that the nature of the toughness transition they observe is qualitatively different from the ductile-brittle transition observed upon varying the temperature. This point is not quite obvious to me. Structural relaxation is known to lower T_f and to embrittle the glass which seems rather consistent with what the authors observe: am I missing something here?
- 4) In line 201, the authors state that $|Y|$ varies gradually with T_f , which is not quite obvious to me, as the last three values (for the highest T_f , see figure 3.d) seem to be on a plateau after a rather significantly steep increase that seems to mimic what's observed in figure 2.a for Kq . However, comparison is made a bit difficult because temperature scales are different between these two figures: could the authors enlighten that point ?
- 5) In line 222, the authors say that strain-rate affects T_f^c , however this is not easily seen in

figure 4.a: the curves at 10⁻⁴ and 10⁻⁵ seem rather superimposed... could the authors give the exact dependency they found by providing the exact value of $T_{fc} = f(\text{strain-rate})$ as they have done in line 131-132 for the three studied compositions ?

6) From line 257 to 265, the authors conclude that thermal excitation has a negligible effect on toughness. This seems rather counter intuitive. They base their conclusion on the values of K_q found at T_{amb} and T_f for a sample with $T_f = 300^\circ\text{C}$. Would it be possible that at such low T_f value, the glass be almost completely relaxed and that the brittle/ductile transition temperature related to thermal excitation has increased closer to T_g ? A full $K_q = f(T)$ curve up to T_g would be necessary, for various T_f (which would be a full other paper in and of itself) to conclude as the authors do, in my opinion. I think it would be best to be more nuanced and point out that this result might suggest that thermal excitation could be negligible, something along these lines.

7) In line 268-270, the authors mention MD studies but do not discuss them : I would suggest either discussing what they have found and how it relates to their finding in the text or remove this part altogether along with all related material.

8) From line 277 to line 282, the authors say that T_g is defined by viscosity reaching a certain value (10^{12} Pa.s). I am not quite sure that this point is so obvious. Either T_g is determined by viscosity measurements and defined as the temperature at which viscosity reaches that value, and then I agree, or it is determined by DSC (which, I imagine, is the case here) and then, experimentally, viscosity appears dependent upon chemical composition at T_g . See for instance [Moynihan, J. Amer. Ceram. Soc., 76 (5) 1081-87, 1993] for inorganic glasses, where viscosity at T_g varies between $10^{10.7}$ and $10^{12.2}$ depending on composition (I have not been able to find similar data for MGs in my rapid bibliography review of that point, but it seems difficult to imagine that the situation be any different in their case). Obviously, if my point is correct, this should modify significantly the authors discussion that follows as T_f^c/T_g ceases to be an a priori universal parameter.

9) A few details about supplementary materials: fig S.1 c should be enlarged and its legend contains many typos.

We would like to thank all reviewers for their insightful and constructive comments. All of the comments have been carefully and seriously considered, a process that has led to almost one year of additional research.

The main outcomes of this process are as follows:

- Additional fracture toughness experiments have been performed in order to better reveal and characterize the observed toughening transition (Fig. 1A).
- An entirely new series of experiments revealed the dependence of the fracture toughness on the ordinary temperature, allowing us to clearly distinguish between the strong dependence of the toughness of the fictive temperature and the much weaker dependence on the ordinary temperature (Fig.1B).
- New experiments spanning a broader range of applied strain rates revealed a similar toughening transition as a function of the strain rate for two different fixed values of the fictive temperature (Fig. 3). These findings significantly strengthen the proposed physical picture.
- The significantly extended experimental data set has led to deeper insight into the physical origin of the toughening transition, to elucidating the differences compared to previously reported brittle-to-ductile transition and to identifying similarities with the ordinary glass transition (Fig. 4).
- As a results of all of the above, except of Fig. 3 of original manuscript, all figures have been either completely changes or extensively revised (Fig. 1)

Finally, the manuscript has been essentially rewritten from scratch, including a new title.

We believe that with these extensive revisions, following the reviewers' insightful comments and suggestions, the revised manuscript clearly reports on new and exciting results, and as such will be judged suitable for the broad readership of Nature Communications.

Below we provide detailed point-by-point responses to the reviewers' comments.

Reviewer #1 (Remarks to the Author):

The authors studied the fictive temperature dependence of fracture toughness of some metallic glasses and linked it to the structural changes. They measured conditional (notch) fracture toughness, KQ for different T_f , and found a characteristic fictive temperature where fracture toughness exhibits non-trivial and abrupt increase from brittle to ductile behavior. They attribute this change in fracture toughness to the competition between two different time scales. They regarded this transition as a mechanical glass transition. I should mention that such transition has been reported in literature also for other glass formers. The results are interesting and has practical implications. The paper is well written.

Our response:

We thank the reviewer for identifying the merits of our work.

In relation to the reviewer's comment that such transition has been reported in literature also for other glass formers, we would like to note that we have not been able to find reports in the literature of a transition similar to the one we report and explain here. There have been reports on ductile-to-brittle transitions as a function of the ordinary temperature [1-3]. These transitions are $\sim 1-2$ orders of magnitude less pronounced compared to the transition reported on in our manuscript. Moreover, and not less important, the toughening transition we have discovered is controlled by the **fictive temperature**, which characterizes the initial structural state of the glass. Indeed, in the revised manuscript (see discussion of the new Fig. 1B in the response to reviewer #2 below), we show that these are fundamentally different phenomena. Finally, we provide a theoretical picture that explains the observed transition, which we have not found elsewhere.

Reviewer #1:

But the discussion is not convincing in several places. I list some specific problems below:

1. In Discussion, the author wrote "we argue that the rather abrupt change in $KQ(T_f)$ emerges from a competition of time scales." Although Fig. 4a shows the different KQ for three different strain rates, the competition of text and τ_{plastic} is not shown. In Ref [59], there are some specific explanations about notch toughness as a function of ξ . But this work involves some speculations and lacks clear evidences.

Our response:

Following the reviewer's comment, and in order to more directly demonstrate the competition of timescales involved in the observed fracture toughness transition, we performed new experiments with two values of the fictive temperature, one above the transition temperature for the strain-rate shown in Fig. 1A of the revised manuscript and one below it. The new results, shown below (Fig. 3 in the revised manuscript), span a wide range of applied strain-rates, 4 orders of magnitude for one

fictive temperature and 5 for the other. In this way, we varied τ_{ext} while keeping τ_{plastic} fixed for each value of the fictive temperature.

The new results clearly show that a similar toughening transition to the one observed for a fixed applied strain-rate as a function of the fictive temperature (see Fig. 1A in the revised manuscript) is observed when the applied strain-rate is varied for a fixed fictive temperature, for two values of the latter. These results lend strong support to the proposed competition of timescales and is fully consistent with the theoretical predictions of Ref. [20] in the revised manuscript (Ref. [59] in the original submission).

The theory developed in Ref. [20], while containing a certain degree of (essential) phenomenology, is one of the only theoretical frameworks (probably the only one) which can describe transient (non-steady-state) rate-dependent elastoplastic deformation and has predicted the presently observed toughening transition before the experiments. This is very uncommon in the field.

Reviewer #1:

2. Detailed description about Fig. 4b is needed to make the figure more understandable.

Our response:

Fig. 4b of the original submission has been taken out in the revised manuscript. Instead, based on the new experiments, we summarize the physical insight and the analogy to the ordinary glass transition in the new Fig. 4.

Reviewer #1:

3. As shown in Fig. 4c-g, the authors performed MD simulations. But the simulation details in Methods part are not clear enough. Besides, there are no explanations about the information in these figures.

The meaning of parameter $P(\Delta r)$ are not mentioned, though it can be interpreted according to last part of Methods.

Our response:

We agree. Consequently, the MD simulations part has been completely removed from the revised manuscript. Instead, we carried out a new series of experiments that revealed the (very weak) dependence of the fracture toughness on the ordinary temperature (Fig. 1B).

Reviewer #1:

4. T_f was regarded as the annealing temperature at which the samples were relaxed with the duration of 1.5τ . The physical meaning of their T_f has not been described. What is the difference between their T_f and enthalpic T_f determined by DSC? It would be helpful if they can compare both T_f s. The DSC methods for determining enthalpic T_f can be found in literature.

Our response:

We have revised the description of T_f and addressed the difference between T_f and the enthalpic T_f as determined by DSC experiments. Please see below the revised text from revised manuscript:

“ T_f – a temperature characterizing the structural degrees of freedom of the glass where it has fallen out-of-equilibrium, and is often referred to as the fictive temperature or glass-transition-upon-cooling, which is different from the calorimetric glass transition temperature, T_g . The difference originates from the pronounced differences in absolute rates; T_g is determined upon heating with rates of typically 20 K/min (0.3 K/s) whereas T_f is maintained by cooling from a temperature where metastable equilibrium has been reached with rates of ~ 6000 K/min (~ 100 K/s).”

Reviewer #1:

5. The authors should provide the references for the dependence of relaxation time on temperature in the Pd-based and Pt-based MGs?

Our response:

We added the references for the temperature dependence of relaxation time in the revised manuscript.

Reviewer #1:

6. The T_g values for all samples should be given in the manuscript.

Our response:

We agree, the T_g values of the three alloys are now reported on in the revised manuscript.

Reviewer #1:

7. The authors should use Kelvin as the unit of the T_f and T_g should be. Only using Kelvin, would T_f/T_g have physical meaning.

Our response:

We agree, all temperatures in the revised manuscript are reported in Kelvin.

Reviewer #1:

8. Only 4 points in Figure 2 were plotted in the Pt-based MG, which is not convincing to support the abrupt changes of the fracture toughness around T_g .

Our response:

We have conducted new experiments to address this comment. In the revised manuscript, we added two new data points, for $T_f/T_g = 0.94$ and 1.08 (the highest and lowest fictive temperatures). These data points required particularly challenging experiments as the relaxation time at $T_f/T_g = 0.94$ approaches several weeks. The 10 experiments that we included (5 per data point) confirmed that the toughening transition agrees well with the other two alloys (Zr- and Pd- BMG).

Reviewer #1:

9. Line 38: molecular liquids should be molecular glasses.

Our response:

We have removed this statement altogether from the revised version.

Reviewer #1:

In summary, this paper is not suitable for Nature Communication regarding its originality and general interest. It might suit a materials science journal.

Our response:

We have thoroughly addressed all of the reviewer's comments and concerns, which have been very insightful, have performed new experiments and have essentially rewritten the manuscript. Together with the revision related to the comments of the other reviewers (see below), which also involved

extensive new experiments, the manuscript has been significantly improved. We believe that the revised manuscript offers compelling and highly novel results, of both fundamental and practical importance, which will be of great interest to the broad readership of Nature Communications.

Reviewer #2 (Remarks to the Author):

The great strength of this work is the quality of the data. As shown in earlier work by some of the authors, they have an experimental method by which the notch fracture toughness (KQ) can be measured with a variability of less than 3%. This is remarkable and useful. The authors have also developed a method by which metallic glass samples can be prepared with controlled values of fictive temperature (T_f).

Our response:

We are happy that the reviewer appreciates our method to precisely measure the fracture toughness, which is essential for fundamental studies on metallic glass processing-property-structure relationships.

Reviewer #2:

The main contribution that the authors claim for this work is that the transition in KQ (from a low value to a high value as T_f is increased) is sharp, not gradual. It is good that the authors have been able to better characterize this transition and to show that it is sharp – but this is certainly not a new or unexpected result.

Our response:

The toughening transition we observed is rather abrupt, but not truly singular (sharp). To the best of our knowledge, such a transition in terms of a well-controlled initial structural state of the glass – characterized by a well-defined fictive temperature – has never been observed before. If the reviewer is aware of available works in the literature that are relevant to our work and have not been properly cited, we would be grateful if such references can be explicitly pointed out for us. There have been reports on ductile to brittle transitions on the ordinary temperature scale [1-3]. These transitions are ~1-2 orders of magnitude less pronounced than the here observed transitions (Fig. 1b). We suggest, and provide in this work a theoretical framework that a ductile to brittle transition should be present on the fictive temperature scale and may, but much weaker, be present on the ordinary temperature scale. We explain this transition by the crossover of relevant time scales. Again, to the best of our knowledge a) such a transition has not been reported and b) we provide a theoretical framework to explain this transition which also predicts a transition, however weak, on the ordinary temperature axis.

Reviewer #2:

The authors are entirely correct (line 145) to draw a distinction between the change in KQ at a critical value of T_f and the classical ductile/brittle transition as a function of temperature. Yet the two behaviors are linked. They both show a transition from brittle fracture to ductile failure as a parameter is changed. That parameter can be internal to the glass (T_f), or external (test temperature or strain rate). The authors are correct to focus on the importance of relative time scales (lines 206-223). But for a given applied strain rate, the transition in failure mode can be considered in terms of stress. At a given strain rate (and with other parameters such as T_f and test temperature fixed) there will be a critical stress for the onset of brittle fracture and a (different) critical stress for the onset of ductile failure. The question, rather simply, is which stress is lower? The lower stress, of course, corresponds to the failure mode that will be observed. As some parameter (for example, test temperature or T_f) is changed, the critical stress values will change and which stress is lower may change. If the identity of the lower stress does change, the failure mode changes, and inevitably there is a rather sharp change in toughness. The point is that transitions in toughness are expected to be sharp, associated with a change in failure mode. Could it be that earlier work suggesting smoother transitions suffered from scatter in the measurements of toughness?

Our response:

Overall, we do not think there is a deep disagreement between the reviewer's picture and the physical picture emerging from our results. In this context, we would like to highlight the following points: (i) The reviewer writes: "The authors are correct to focus on the importance of relative time scales (lines 206-223)". We are very happy about this statement, we just want to note that these ideas have not been discussed and experimentally supported in the existing literature. (ii) The reviewer writes: "But for a given applied strain rate, the transition in failure mode can be considered in terms of stress". We agree that a change in the failure mode is involved, as we explicitly show in the fractographic data presented in Fig. 1D of the revised manuscript. Yet the transition is not exactly sharp and more importantly the challenge is to understand the origin of the transition, which is precisely what we offer. Most notably, we show that the initial structural state of the glass, quantified by the well-defined fictive temperature, controls the transition. Within proposed theoretical framework, the fictive temperature is taken to determine the initial population of STZs and consequently to control the plastic relaxation time. The strong dependence on the fictive temperature, predicts a rather abrupt transition in the toughness when the externally applied strain-rate is taken into account. This physical picture is significantly supported by Fig. 3 in the revised manuscript, where the corresponding toughening transition is observed as a function of the strain-rate for a fixed fictive temperature, see discussion above in the response to reviewer #1. Moreover, by performing extensive new experiments in which the ordinary temperature is varied for a fixed fictive temperature, we found essentially no variation in the toughness from T_g down to $0.5 T_g$, see revised Fig. 1B:

In fact, only at the liquid Nitrogen temperature larger changes in the toughness are observed, clearly demonstrating a dramatically higher sensitivity of the toughness to the structural state of the glass quantified by the fictive temperature. (iii) Finally, the reviewer asks: “Could it be that earlier work suggesting smoother transitions suffered from scatter in the measurements of toughness?”. We believe, and explicitly state in the revised manuscript, that previous measurements were not fully reliable as the fictive temperature has not been carefully controlled. In fact, this might have led to spurious transitions (not to smoother behaviors).

Reviewer #2:

In lines 173 and 174, the authors note that “none of these quantities exhibit a T_f dependence similar to that of K_Q ”. This point is reinforced in lines 202 and 203 – but there is nothing remarkable about this. The various quantities (G/B, enthalpy recovery, and static atomic pair correlation function and its response to stress) all show (according to the authors’ own data, Fig. 3) a smooth correlation with T_f . Thus any of these quantities could be used as a proxy for T_f (or T_f/T_g) in plotting a figure such as Fig. 2a. And, to repeat the point made in the previous paragraph, it is straightforward to understand why the toughness shows a sharp transition as a function of any of these variables.

Our response:

The importance of these data, now appearing in Fig. 2 of the revised manuscript, is two-fold: first, they show that the toughness, which is determined by the susceptibility to plastic deformation, varies in a much more dramatic manner w.r.t the fictive temperature compared to these other physical quantities (which are also affected by the glass structure) (ii) It puts our results in the proper literature context, where some of these quantities, e.g. G/B, were claimed to control the toughness, while they are in fact merely correlated to it.

Reviewer #2:

In the paragraph starting on line 257, the authors consider “enthalpic contributions” to KQ. This approach is at risk of confusing thermodynamic and kinetic aspects. As the authors themselves note, there are potential contributions from thermal activation, so that kinetics could be important. The particular example cited by the authors is interesting, but it would be misleading to assume that it is general. Specifically, the authors find that KQ for a glass with $T_f = 300$ degC and measured at 300 degC is the same as for a glass with $T_f = 300$ deg C measured at room temperature. This does not, however, say that the test temperature is unimportant (which, I think, is what the authors are implying when they discount the influence of Hvolume). (This is also implied in the comment about “Enthalpic changes” in lines 306 and 307.)

A glass with a given structure (and given T_f) will have a ductile/brittle transition temperature. Well above or below that temperature, KQ is likely to be rather constant, independent of temperature. Thus the similar values of KQ noted in the previous paragraph simply tell us that room temperature and 300 degC are on the same side of the transition temperature; they do not imply that, overall, “contributions from volume expansion and thermal excitation are negligible” (line 265).

Our response:

We have removed altogether the discussion “enthalpic contributions” from the revised manuscript. In addition, as discussed above, we have included complete new measurement and discussion of the effect of the ordinary temperature, showing that the fracture toughness is far less sensitive to it compared to the fictive temperature.

Reviewer #2:

On line 302, the authors state that a transition from brittle-like to ductile-like behavior “is absent in conventional models”. This statement seems too strong, given the overall interest in transitions in the failure mode.

Our response:

We have removed this statement altogether from the revised manuscript.

Reviewer #2:

Minor points:

In line 134, the authors discuss “KQ at T_g ”. That would normally mean ‘KQ measured at T_g ’, which is not what the authors intend. Instead, they mean something like: “KQ at $T_f=T_g$ ”.

In Fig. 2b, the images may be too small to be useful.

Line 331: should be 'Tammann'

Our response:

These problems have been fixed.

Reviewer #2:

Overall -- really good data giving a comprehensive characterization of the ductile/brittle transition, but not offering new physical insights.

Our response:

We thank the reviewer for identifying the merits of our work and for his/her insightful and constructive comments which have helped significantly in improving this work and making the new physical insight more assessable. We believe that the revised manuscript not only presents new data that even further strengthen the experimental aspects of our work, but that these data also provide strong evidence in favor of the suggested physical picture. The latter is further highlighted in the revised manuscript by the new Fig. 4.

The essentials of new physical insights are:

- Observed ductile to brittle transition on the fictive temperature axis originates from a crossover of time scales
- Involved time scales are extrinsic (loading strain rate) and intrinsic (plastic dissipation rate).
- As τ_{plastic} is a much stronger function of T_f than of T , the transition of $K_q(T_f)$ is much more pronounced than $K_q(T)$, which we have also experimentally confirmed.
- The transition temperature is a function of strain rate, which is predicted from our theoretical framework and we also show this experimentally (Fig. 3 in revised manuscript)

As such, we believe that the revised manuscript offers compelling and highly novel results, of both fundamental and practical significance, which will be of great interest to the broad readership of Nature Communications.

Reviewer #3 (Remarks to the Author):

The paper proposed by Ketkaew et al. on the control of ductility in MG by fictive temperature is, in my opinion, definitely worthy of publication in Nature. Indeed, it establishes for the first time a rather revolutionary and fundamental fact, based on solid experimental - and original - results along with interesting discussion: the fictive temperature control the glass toughness. It opens new and very wide opportunities for academic research in all of glass science (a priori not limited to metallic glasses) and will have important implications for the use of BMG as structural materials, in particular concerning the heat treatments that should be applied to improve their toughness.

The main finding of the paper is that MG ductility is strongly controlled by their fictive temperature, in a very systematic way, on three different compositions (Pt, Pd and Zr based MG) with a particular care taken about the meaningfulness of the results (which are definitely meaningful: no experimental error seems able to account in any way for the results which are extremely consistent). This finding has never been reported elsewhere, to the best of my knowledge, and this very question is addressed for the first time, as the experimental control is tricky and needs the great experience the authors have. What's more, the discussion is based on additional structural analysis that opens fundamental implications of prime importance in the field (notably the variation with temperature of G/B , recovery enthalpy, $G(rp)$ and $|Y|$ and their correlation with Kq).

Our response:

We are grateful to the reviewer for these very encouraging comments and for the recommendation to publish the manuscript in Nature Communication.

Reviewer #3:

The main claims are definitely convincing, but some aspect could, in my opinion, be improved or nuanced without great effort, I'd say, as it would not involve any additional experimental work. So, however superb the results and the work, I do have a few points which, for some of them, seem significant, and also a few questions. I present my points as the text goes, not by order of importance:

1) In line 85 and 86, the authors state that they "provide experimental evidence that upon cooling from T_f to room temperature [...] the glass remains in a single 'meta basin' in the energy landscape." : I must say that this point is actually not made clearly enough in the text (or I may have missed it) and it would be worth being made clearer because of its importance.

Our response:

We have removed this statement altogether from the revised manuscript.

Reviewer #3:

2) Line 136-139, the authors mention that for ZrTiCuNiBe, the fracture pattern evolves significantly and

they show said evolution in figure 2.c. It would be worth at least discussing what they have observed for the other two glasses, and possibly include SEM images (if available) in the supplementary material. On the side note, figure 2.c should be made larger for comfort of viewing.

Our response:

We completely agree. In the revised supplementary we have now a discussion about the various fracture surfaces and have added SEM images of the Pd and Pt BMG.

Reviewer #3:

3) The authors stress that the nature of the toughness transition they observe is qualitatively different from the ductile-brittle transition observed upon varying the temperature. This point is not quite obvious to me. Structural relaxation is known to lower T_f and to embrittle the glass which seems rather consistent with what the authors observe: am I missing something here?

Our response:

We have performed extensive new experiments to elucidate the relation between varying the fictive temperature and the ordinary temperature, as explained above in the response to reviewer #2. In the revised Fig. 1B:

we clearly show that the fracture toughness is far less sensitive to variations in the ordinary temperature than to the fictive temperature (by comparison to Fig. 1A). Indeed, for very large reduction in the ordinary temperature, down the liquid Nitrogen temperature, the toughness eventually starts to drop.

Reviewer #3:

4) In line 201, the authors state that $|Y|$ varies gradually with T_f , which is not quite obvious to me, as the last three values (for the highest T_f , see figure 3.d) seem to be on a plateau after a rather significantly steep increase that seems to mimic what's observed in figure 2.a for K_q . However, comparison is made a bit difficult because temperature scales are different between these two figures: could the authors enlighten that point?

Our response:

The impression that the reviewer has is related to the fact that a **single data point**, the one at $T_f/T_g=1$, appear not to follow the very smooth dependence of $|Y|$ on T_f . Most likely, there has been a systematic error in this particular measurement, but we do not think it changes the fact that the function exhibits a smooth and gradual variation with T_f . Indeed, no corresponding behavior is observed in $G(r)$.

Reviewer #3:

5) In line 222, the authors say that strain-rate affects T_f^c , however this is not easily seen in figure 4.a: the curves at 10^{-4} and 10^{-5} seem rather superimposed... could the authors give the exact dependency they found by providing the exact value of $T_{fc} = f(\text{strain-rate})$ as they have done in line 131-132 for the three studied compositions?

Our response:

As we explained in detail above (please see the response to reviewer #1), we have significantly revised this the part that addresses the strain-rate dependence and added new experimental data in the revised Fig. 3:

This figure explicitly establish the complementary observation that the corresponding transition in terms of the applied strain-rate is fictive temperature dependent.

Reviewer #3:

6) From line 257 to 265, the authors conclude that thermal excitation has a negligible effect on toughness. This seems rather counter intuitive. They base their conclusion on the values of Kq found at T_{amb} and T_f for a sample with $T_f = 300^\circ\text{C}$. Would it be possible that at such low T_f value, the glass be almost completely relaxed and that the brittle/ductile transition temperature related to thermal excitation has increased closer to T_g ? A full $Kq = f(T)$ curve up to T_g would be necessary, for various T_f (which would be a full other paper in and of itself) to conclude as the authors do, in my opinion. I think it would be best to be more nuanced and point out that this result might suggest that thermal excitation could be negligible, something along these lines.

Our response:

As explained in detail above, the effect of ordinary thermal fluctuation on the toughness is now extensively addressed in the revised manuscript.

Reviewer #3:

7) In line 268-270, the authors mention MD studies but do not discuss them: I would suggest either discussing what they have found and how it relates to their finding in the text or remove this part altogether along with all related material.

Our response:

We followed the reviewer's advice and removed the MD simulation part altogether.

8) From line 277 to line 282, the authors say that T_g is defined by viscosity reaching a certain value (10^{12} Pa.s). I am not quite sure that this point is so obvious. Either T_g is determined by viscosity measurements and defined as the temperature at which viscosity reaches that value, and then I agree, or it is determined by DSC (which, I imagine, is the case here) and then, experimentally, viscosity appears dependent upon chemical composition at T_g . See for instance [Moynihan, J. Amer. Ceram. Soc., 76 (5) 1081-87, 1993] for inorganic glasses, where viscosity at T_g varies between $10^{10.7}$ and $10^{12.2}$ depending on composition (I have not been able to find similar data for MGs in my rapid bibliography review of that point, but it seems difficult to imagine that the situation be any different in their case). Obviously, if my point is correct, this should modify significantly the authors discussion that follows as T_f^c/T_g ceases to be an a priori universal parameter.

Our response:

We agree with the reviewer that quantifying T_g has been somewhat ambiguous. Within the metallic glass community, T_g is considered as the temperature of the endothermic reaction when heating a glass at 20K/min. The viscosities associated with T_g are very close to 10^{12} Pa s [4,5].

Reviewer #3:

9) A few details about supplementary materials: fig S.1 c should be enlarged and its legend contains many typos.

Our response:

We agree. We enlarge fond and eliminated all typos. In addition, we added the SEM images of fracture surfaces for the Pd and Pt BMG.

In summary, through addressing the reviewers constructive and insightful comments and carry out additional experiments, we believe that the revised manuscript conveys the finding and explanation of a mechanical glass transition better and assessable to a broad community.

References:

1. Gu, X.J., S.J. Poon, G.J. Shiflet, and J.J. Lewandowski, *Ductile-to-brittle transition in a Ti-based bulk metallic glass*. Scripta Materialia, **60**, 1027-1030, (2009).
2. Raghavan, R., P. Murali, and U. Ramamurty, *On factors influencing the ductile-to-brittle transition in a bulk metallic glass*. Acta Materialia, **57**, 3332-3340, (2009).
3. Barth, E.P., F. Spaepen, R. Bye, and S.K. Das, *Influence of processing on the ductile-to-brittle transition temperature of an Fe-B-Si metallic glass*. Acta Materialia, **45**, 423-428, (1997).
4. Busch, R., J. Schroers, and W.H. Wang, *Thermodynamics and kinetics of bulk metallic glass*. Mrs Bulletin, **32**, 620-623, (2007).
5. Busch, R., *The thermophysical properties of bulk metallic glass-forming liquids*. Jom-Journal of the Minerals Metals & Materials Society, **52**, 39-42, (2000).

Reviewers' comments:

Reviewer #1 (Remarks to the Author):

The revised paper has been improved by considering the review reports. The authors explored the evolution of KQ with T_f for three metallic glasses, and found a sharp brittle-to-ductile transition upon increasing T_f , arising from the competition between plastic deformation time scale and loading time scale. In addition, the authors identified mechanical glass transition by using the threshold fictive temperature, T_{fDB} . As I can see, the fracture toughness data were accurate and reliable. However, I still feel that several points in the paper are confusing and not convincing and need to be addressed in more detail. I suggest the authors to consider the following points:

Major issues:

1. Line 109: '~6000 K/min (~100 K/s)'.

I wonder how the authors obtained the cooling rate of ~100 K/s. It is important to ensure sufficiently high cooling rate to get the glass frozen-in at the annealing temperature, and thereby to obtain the targeted T_f . Otherwise, the T_f would be lower than the annealing temperature. Therefore it is crucial to precisely calculate the real T_f value (see Y.Z. Yue, *J. Non-Cryst. Solids* 354 (2008) 1112. In other words, DSC measurements could be used to verify that the annealed glass indeed fall out of equilibrium at the annealing temperature. Since the cooling rate can vary in several orders of magnitude, T_f can change significantly. At least, one DSC measurement should be done as an example to show their cooling rate is sufficient.

2. T_f can be determined in different ways, e.g., by thermal or mechanical approaches, as described in A.Q. Tool, *J. Am. Ceram. Soc.* 29 (1946) 240-253; O. S. Narayanaswamy, *J. Am. Ceram. Soc.* 54 (1971) 491-498. The authors should mention these papers which introduced the concept of fictive temperature.

3. The authors mentioned that the DB transition as a function of T_f is fundamentally different from that as a function of ordinary temperature. This is the main finding of this paper. To substantiate their arguments, the authors could address the following points:

1) What is the temperature at which the fracture toughness was measured (red data points in Fig. 1b)? Room temperature (RT) or ordinary temperature T ? I cannot find it neither in the protocol (lines 89-100) nor in method section (lines 349-368). However, I guess that the temperature of measurement is RT. Therefore, your red data points in Fig. 1b are different from those reported in literature since the latter were obtained at different temperatures from yours. Thus it is hard to say that your DB transition is different from that reported in literature.

2) When the ordinary temperature T is varied, the cooling rate would be also changed if the same cooling technique is used. The change of T from 77 K to 573 K won't change T_f too much, because at T below T_f , the glass is already frozen in, no configurational change occurs. Therefore, KQ certainly won't significantly vary with T , and hence the change of KQ with T (red data points in Fig. 1b) is less than that of KQ with T_f (black data points in Fig. 1b).

4. The T_f dependence of KQ is interesting. However, the effect of T_f on glass mechanical properties has been reported in literature, e.g., M.M. Smedskjaer, et al., *J. Non-Cryst. Solids* 356 (2010) 893-897, where it is reported that the hardness decreases (equivalent to increase of fracture toughness) with increasing T_f . Those studies should be mentioned.

Minor issues:

1. Lines 97-98.

Does τ_{SR} here mean $\tau_{SR}(T_f)$ or $\tau_{SR}(T_g)$ or the average value of τ_{SR} from T_g to T_f ? It should be specified because the authors stated that T_f varied by ~100 K, corresponding to more than 6 orders of magnitude change in τ_{SR} , leading to a large change of quenching rate.

2. Lines 149-150, 'in the transition region ...'

How did you obtain a threshold value of T_{fDB} if the transition happens in a region? "transition region" might not be a proper term.

3. Fig. 1 c, d: the second sample, $T_f=603$ K, T_f/T_g should be 0.968, not 0.997.

4. Fig. 1 caption: 1a (line 171), there is no inset in the figure; 1b, please specify the value of T_f for red circles and triangles data

5. Lines 244-247: I think you mean 'As the local stresses and T are identical ...'

In summary, the paper should be revised by considering the above-mentioned aspects.

Reviewer #2 (Remarks to the Author):

This manuscript in its revised form retains the good features of the original version. These are the care with which the samples have been prepared to achieve well characterized values of the fictive temperature T_f , and the quality of the data on notch fracture toughness KQ . With these data of unusually high quality, the ductile-to-brittle transition (DBT) in metallic glasses (three in this study) can be examined in detail.

The authors make several broad claims and attempt (as stated in the title of the paper) to present the DBT as a "mechanical glass transition". The increase in insight into the DBT is not as great as the authors claim. These concerns are explained further below.

It is worth noting that there is a literature on the DBT in metallic glasses, notably the paper by Wu & Spaepen [Philos. Mag. 61 (1990) 739–750] (cited as Ref. [15] in the manuscript). This suggests a classical picture in which (for a metallic glass in a given state) there is a sharp step-like transition to lower toughness as the temperature is lowered. The temperature of this step (the ductile-to-brittle transition temperature DBTT) increases as the glass is relaxed by annealing. For a given test temperature, the DBTT of the as-cast glass could be below the test temperature, but evolve to a value higher than the test temperature as a result of annealing. Thus a DBT would be observed at the given test temperature as a function of the degree of annealing.

The discussion in lines 104 to 109 is confusing. It compares the glass-transition temperature T_g with the fictive temperature T_f , and states that "the difference originates from the pronounced differences in the absolute rates". However, the value of T_f is set not by the cooling rate, but by the annealing temperature used by the authors. The cooling rate does need to be high enough to preserve the state of the glass, but the cooling rate does not determine T_f . And the difference with the heating rate used to determine the conventional T_g is not really relevant. Heating in the DSC at the conventional rate of 20 K/min is capable of determining values of T_f as well as T_g .

In lines 132 to 133, the authors claim that a transition in toughness as a function of T_f has not previously been reported. This may strictly be true, but the authors' finding is not as novel as this statement suggests. The work by Wu & Spaepen (cited above) shows the variation of toughness with annealing, and links the DBTT with the enthalpy of the glass. The enthalpy could easily be linked to T_f .

Since (lines 300 to 301) the authors claim to have developed a procedure "to significantly enhance the fracture toughness of metallic glasses", it is strange that they do not compare the KQ of their treated samples with the KQ of the freshly thermoplastically formed samples. It is true that the properties of the formed samples may show scatter, but it nevertheless would be good to see a quantitative demonstration of toughening.

In lines 160 to 162, the authors note that "surprisingly" KQ has a much stronger dependence on T_f than on the test temperature T . But this is really not surprising. As the authors themselves have noted (lines 99 to 100), there is a variation of more than six orders of magnitude in relaxation times corresponding to the range of T_f . The variation of relaxation time is also reflected in the strong dependence of the liquid viscosity on temperature as the glass transition is approached on cooling. Because the glass (unlike the liquid) is isoconfigurational, the dependence of its viscosity on temperature is much weaker than that of the liquid. Correspondingly, the variation of KQ (or other properties) with test temperature can be rather weak. In other words, the variation of KQ

with T_f reflects the temperature dependence of liquid properties, the variation of KQ with T reflects the temperature dependence of glass properties.

In lines 185 to 206, the authors contrast the smooth variation of most properties as a function of T_f with the sharp change seen for KQ . However, the DBT reflects a change in fracture mode and always appears as a step-like change, even though the underlying properties (e.g. the relaxation time in the glass) vary smoothly.

In this work, the authors control the glass structure by annealing (characterized by changes in T_f). In lines 299 to 315, they contrast this type of control with that achieved by varying the cooling rate through the glass transition. In principle, both methods give access to a range of glassy states and there is no fundamental reason why one should correspond to sharp and one to gradual changes in KQ , as would be evident if the same parameter (e.g. T_f) were used to describe the range of glassy states in each case. The authors are certainly correct that the variation of cooling rate may introduce significant scatter, while annealing gives well defined, reproducible states.

As already noted, the authors claim that their controlled annealing is a procedure “to significantly enhance the fracture toughness of metallic glasses”. This is potentially misleading. In general, annealing leads towards structural relaxation and embrittlement. So the claimed “toughening transition” (line 154) can in some cases simply be a transition to a state that has undergone less relaxation (in other words a transition towards the as-cast state).

The thermoplastically formed samples may be sufficiently relaxed that the authors’ high- T_f treatments can reduce the degree of relaxation and therefore toughen the samples. As noted above, no experimental evidence has been given for this. Reducing the degree of relaxation by thermal treatments relies on having a high enough cooling rate and has been quantitatively demonstrated by others – Wakeda et al., Controlled rejuvenation of amorphous metals with thermal processing, *Scientific Reports* 5 (2015) 10545.

In Fig. 4 and the associated discussion, the authors attempt to demonstrate a degree of correspondence between the conventional glass transition and the DBT, which is likened to a mechanical glass transition. This may be striving too hard to demonstrate some new fundamental insight that is not there. In particular, the attempts to make Figs 4a, 4b and 4c look similar to each other distorts observed behavior and is misleading.

In the conventional glass transition (Fig. 4a), the property of the glass lies ABOVE the extrapolated liquid line at the same temperature – i.e. in terms of the liquid line, the glass has properties at a given temperature that reflect an effective temperature that is higher than the actual temperature. For the claimed mechanical glass transition, however (Figs 4b and 4c), the reverse is true – which clearly reflects different behavior.

In Fig. 4a the enthalpy line for the glass is shown curved, presumably to make an analogy with the toughness lines for the glass in Figs 4b and 4c. But, for actual glasses, the enthalpy line is not noticeably curved. The key problem is that the DBT depicted as toughness versus temperature show step-like behavior, quite distinct from the ‘change of slope’ behavior of the glass transition. If, in Fig. 4a, the authors were to plot specific heat capacity rather than enthalpy, then a better analogy (at least superficially) would emerge.

In the caption to Fig. 1a, there is mention of an inset, but there is no inset in the figure.

Regarding the responses to reviewers’ comments:

Reviewer 1, point 1:

The dispute is over whether such a transition has been previously reported or not. As noted above, earlier work on annealing effects on toughness (Wu & Spaepen) and on rejuvenating effects of

thermal treatments (Wakeda et al.) have covered much of this ground (but without describing the glassy state in terms of T_f).

Reviewer 1, point 5:

This point has not been addressed by the authors. The T_f of any sample can be assessed by DSC measurements, and it would indeed be interesting to compare this with the nominal T_f corresponding to the thermal treatment.

Reviewer 2, point 2:

The authors' approach of describing the changes in the glasses in terms of T_f is attractive, but there is earlier work (e.g. Wu & Spaepen) showing (less quantitatively) many of the same effects.

Reviewer #3 (Remarks to the Author):

The - entirely - revised version of the paper proposed by Ketkaew et al., titled Mechanical glass transition revealed by the fracture toughness of metallic glasses, provides new great insights of fundamental as well as practical importance.

All the commentary I had made for the first version have been answered, and beyond.

I see only a few points to make:

1) Line 160-162, the authors state that the variations of K_q with T_f are 100 times larger than the variation of K_q with T . I don't see how that follows from their data. They write : "100 times larger than the negligible variation of K_q with T over the same temperature range" : this expression should be clearly justified. What range are the authors considering and what are their reasons to consider this range particularly ? Depending on the considered range, fig 1.b might show no sign of a 100 ratio between the variation of K_q with T with respect to its variation with T_f . Indeed, for high T_f materials, the measured variation of K_q is approximately $110-60 = 50$, whereas the DB variation with T_f is approximately $110-40 = 70$, which is at best a factor 2 ratio. It's just that the BDT controlled by T or T_f occur at different temperatures and the control of ductility by one parameter or another (T or T_f) should be made between the highest and lowest value of K_q .

2) In spite of the great care brought to experimental measurements, the rather small number of data points in the cryogenic part of fig 1.b does not permit to know whether the whole thermal ductile-brittle transition has been captured. If I had to take a bet, my guess would be that no and that measurements at even lower temperatures might reveal that the high T_f curve (circles) converges with the low T_f (triangles) in fig 1.b. But that is just a guess and maybe the authors have additional data or information on that particular point. I don't think additional experiments are absolutely necessary to verify this, though, as it is not the center of the present study, but I think it should be included in the discussion to mitigate the conclusions. As a matter of fact, it is impossible to say, from these data (3 points), if the whole thermal BDT has been measured and conclusions cannot be made too positively on that particular point.

3) In a very rough manner, I think it would more appropriate to consider that T_f seems to control an athermal or structural value of K_q in the ductile regime (so to speak) and that there is still a thermal ductile brittle transition, as the data presented by the authors seem to suggest. This point of view might have also implication on the 100 ratio discussion in point 1. Moreover, it seems to me that T_{BDT} appears independent of T_f , still from fig 1.b, which could be made explicit (in a short manner, perhaps) if this seems correct to the authors.

Beyond these 3 points, I think this paper of really great value, I am unaware of any similar results

having ever being published (or I'd like to be clearly pointed to them if they exist) and I highly recommend its publication in Nature Comm.

We thank the reviewers for carefully reading the manuscript, and for their constructive comments and suggestions to improve the manuscript. Please find below in blue our detailed, point-to-point responses to the reviewers' comments.

Reviewer #1:

The revised paper has been improved by considering the review reports. The authors explored the evolution of KQ with T_f for three metallic glasses, and found a sharp brittle-to-ductile transition upon increasing T_f , arising from the competition between plastic deformation time scale and loading time scale. In addition, the authors identified mechanical glass transition by using the threshold fictive temperature, T_{fDB} . As I can see, the fracture toughness data were accurate and reliable. However, I still feel that several points in the paper are confusing and not convincing and need to be addressed in more detail. I suggest the authors to consider the following points:

We thank the reviewer for these comments. We addressed the reviewer's points in detail below and made the corresponding changes in the revised manuscript.

Major issues:

1. Line 109: '~6000 K/min (~100 K/s)'. I wonder how the authors obtained the cooling rate of ~100 K/s. It is important to ensure sufficiently high cooling rate to get the glass frozen-in at the annealing temperature, and thereby to obtain the targeted T_f . Otherwise, the T_f would be lower than the annealing temperature. Therefore it is crucial to precisely calculate the real T_f value (see Y.Z. Yue, *J. Non-Cryst. Solids* 354 (2008) 1112. In other words, DSC measurements could be used to verify that the annealed glass indeed fall out of equilibrium at the annealing temperature. Since the cooling rate can vary in several orders of magnitude, T_f can change significantly. At least, one DSC measurement should be done as an example to show their cooling rate is sufficient.

High cooling rates have been achieved in our experiments by quenching the samples after annealing at a prescribed temperature T_f into water. The emerging cooling rate of ~100 K/s has been estimated by two independent methods:

- (a) First, we inserted a thermocouple into a sample and directly measured ~100 K/sec for temperatures from annealing temperature to T_g .*
- (b) Second, the cooling rate has been analytically estimated by solving the heat equation using $T_{max} \sim 673K$, $T_{min} \sim 300K$ (water temperature in most experiments), a sample thickness of ~0.7mm, width of ~10mm, and a thermal diffusivity of ~2 mm²/s, representative of the metallic glasses under consideration. Under these conditions, cooling times of a few seconds have been obtained, resulting in an average cooling rate of ~100K/s (the temperature change is $T_{max} - T_{min}$). This internal consistency allowed us to confidently report on a cooling rate of ~100 K/s.*

*In addition, following the reviewer's suggestion, we performed heat capacity measurement according to [Yue Y-Z. Characteristic temperatures of enthalpy relaxation in glass. *Journal of Non-Crystalline Solids* 354, 1112-1118 (2008). And Yue YZ, Christiansen Jd,*

Jensen SL. Determination of the fictive temperature for a hyperquenched glass.
Chemical Physics Letters **357**, 20-24 (2002).

] to demonstrate that indeed our cooling rate is sufficiently large to render T_f upon cooling as the annealing temperature. We performed heat capacity measurement for $Zr_{44}Ti_{11}Cu_{10}Ni_{10}Be_{25}$ prepared through equilibrating (annealing) at 683K followed by rapid quenching to obtain a fictive temperature of 683K as described in the manuscript. The determined fictive temperature through heat capacity measurement where we equated the recovered enthalpy with the excess inherent structural energy. We determined identical fictive temperature through this measurement which verified that the fictive temperature is 683K (see figure below for experimental result). We also used the same technique on sample equilibrated at 693K which allow us to conclude that at 693 K our protocol is insufficient to maintain the annealing temperature as the fictive temperature. We have been unaware of this method and considering it now to use it to study other metallic glass related phenomena outside the focus of this work. We added a section in the supplementary material to describe this measurement as a method to determine the fictive temperature. We also refer to this method in the main text.

2. T_f can be determined in different ways, e.g., by thermal or mechanical approaches, as described in A.Q. Tool, J. Am. Ceram. Soc. 29 (1946) 240-253; O. S. Narayanaswamy, J. Am. Ceram. Soc. 54 (1971) 491-498. The authors should mention these papers which introduced the concept of fictive temperature.

We added references to the classical papers of Tool and Narayanaswamy to the revised manuscript.

3. The authors mentioned that the DB transition as a function of T_f is fundamentally different from that as a function of ordinary temperature. This is the main finding of this paper. To substantiate their arguments, the authors could address the following points:

1) What is the temperature at which the fracture toughness was measured (red data points in Fig. 1b)? Room temperature (RT) or ordinary temperature T ? I cannot find it neither in the protocol (lines 89-100) nor in method section (lines 349-368). However, I guess that the temperature of measurement is RT. Therefore, your red data points in Fig. 1b are different from those reported in literature since the latter were obtained at different temperatures from yours. Thus it is hard to say that your DB transition is different from that reported in literature.

The reviewer refers to the red data points (both circles and triangles) in Fig. 1b. These data have been obtained for two fixed values of T_f and variable test temperature's T . The reference data (black squares) have been characterized at RT for variable T_f (see panel a). To further clarify this, we added the red text to the caption, which now reads: " K_Q for $Zr_{44}Ti_{11}Ni_{10}Cu_{10}Be_{25}$ as a function of T_f (measured at room temperature, black symbols – bottom axis) and T (red symbols – top axis, measured at T) with $T_f = 683K > T_f^{DB}$ (red circles) and $T_f = 583K < T_f^{DB}$ (red triangles)".

In fact, in the next comment, the reviewer properly interpret the presented data: "...and hence the change of K_Q with T (red data points in Fig. 1b) is less than that of K_Q with T_f (black data points in Fig. 1b)", which is exactly what is shown in Fig. 1b (and explained again in the previous paragraph). We trust that this clarifies this major point, and also explains why these data are directly comparable to those previously reported in the literature.

2) When the ordinary temperature T is varied, the cooling rate would be also changed if the same cooling technique is used. The change of T from 77 K to 573 K won't change T_f too much, because at T below T_f , the glass is already frozen in, no configurational change occurs. Therefore, K_Q certainly won't significantly vary with T , and hence the change of K_Q with T (red data points in Fig. 1b) is less than that of K_Q with T_f (black data points in Fig. 1b).

We agree that changing T also affects the cooling rate. Yet, for our experiments, the maximal variation is ~50%, so no significant effects are expected. We also agree that changes in T below T_f do not lead to significant configurational changes, which is exactly what allows us to separate the effect of T_f from that of T , and directly demonstrate the difference between the variations of the toughness with respect to each of them.

4. The T_f dependence of K_Q is interesting. However, the effect of T_f on glass mechanical properties has been reported in literature, e.g., M.M. Smedskjaer, et al., *J. Non-Cryst. Solids* 356 (2010) 893-897, where it is reported that the hardness decreases (equivalent to increase of fracture toughness) with increasing T_f . Those studies should be mentioned.

We refer to the work of Smedskjaer et al., which we were not aware of, in the revised manuscript and thank the reviewer for pointing it out to us. We stress, though, that in the work of Smedskjaer et al., as in several other previous works, the reported variation in the hardness is gradual. Our main discovery, on the other hand, is that the fracture toughness exhibits a rather abrupt transition under similar conditions which originates from a cross-over of involved time scales.

Minor issues:

1. Lines 97-98.

Does τ_{SR} here mean $\tau_{SR}(T_f)$ or $\tau_{SR}(T_g)$ or the average value of τ_{SR} from T_g to T_f ? It should be specified because the authors stated that T_f varied by ~ 100 K, corresponding to more than 6 orders of magnitude change in τ_{SR} , leading to a large change of quenching rate.

τ_{SR} presents the relaxation time at T_f , as implied by "...the structural relaxation time, τ_{SR} , at that temperature". To make it more clear, we changed the text to read "...the structural relaxation time, τ_{SR} , at T_f ".

2. Lines 149-150, 'in the transition region ...'

How did you obtain a threshold value of T_{fDB} if the transition happens in a region? "transition region" might not be a proper term.

The reviewer is right, we modified "in the transition region" to "at the transition". The transition is rather abrupt, not infinitely sharp, so the exact value of T_f^{DB} features a small uncertainty, however, has no implications on our findings and interpretations.

3. Fig. 1 c, d: the second sample, $T_f=603$ K, T_f/T_g should be 0.968, not 0.997.

4. Fig. 1 caption: 1a (line 171), there is no inset in the figure; 1b, please specify the value of T_f for red circles and triangles data

5. Lines 244-247: I think you mean 'As the local stresses and T are identical ...'

2. Correct, we have modified revised manuscript accordingly.
3. Correct, we have revised the figure caption and we added the T_f values for red circles and triangles in the figure caption.
4. Correct, we have revised manuscript accordingly.

In summary, the paper should be revised by considering the above-mentioned aspects.

We have addressed all of the reviewer's comments and suggestions, and have made the corresponding changes in the revised manuscript. We thank the reviewer for the constructive approach and for helping us improving the manuscript.

Reviewer #2:

This manuscript in its revised form retains the good features of the original version. These are the care with which the samples have been prepared to achieve well characterized values of the fictive temperature T_f , and the quality of the data on notch fracture toughness K_Q . With these data of unusually high quality, the ductile-to-brittle transition (DBT) in metallic glasses (three in this study) can be examined in detail.

We thank the reviewer for identifying the unusual quality of our results.

The authors make several broad claims and attempt (as stated in the title of the paper) to present the DBT as a “mechanical glass transition”. The increase in insight into the DBT is not as great as the authors claim. These concerns are explained further below.

The reviewer’s concerns are addressed in detail below.

It is worth noting that there is a literature on the DBT in metallic glasses, notably the paper by Wu & Spaepen [Philos. Mag. 61 (1990) 739–750] (cited as Ref. [15] in the manuscript). This suggests a classical picture in which (for a metallic glass in a given state) there is a sharp step-like transition to lower toughness as the temperature is lowered. The temperature of this step (the ductile-to-brittle transition temperature DBTT) increases as the glass is relaxed by annealing. For a given test temperature, the DBTT of the as-cast glass could be below the test temperature, but evolve to a value higher than the test temperature as a result of annealing. Thus a DBT would be observed at the given test temperature as a function of the degree of annealing.

We have extensively discussed the Wu-Spaepen work during the project, and even personally contacted Frans Spaepen to discuss with him our results (this is the origin of the personal acknowledgement to him at the end of the manuscript). As the reviewer notes, we also cite the paper and refer to it in the manuscript. We appreciate this work and its role in the development of the field.

In our view, the main contribution of the Wu-Spaepen work is that it demonstrated that structural relaxation also contributes to embrittlement. However, Wu and Spaepen – and in fact most of the community up until now – thought that the ordinary (test) temperature, NOT the fictive temperature, is controlling the DBT in these materials. Indeed, essentially all previous work on the DBT in metallic glasses is attributed to changes in the ordinary (test) temperature. Wu and Spaepen showed that annealing affects this transition, i.e. it can make it more pronounced and can shift its location. Yet, the initial structural state of the glasses employed in the Wu-Spaepen work has not been carefully controlled, i.e. has not been characterized by a well-defined fictive temperature, and the whole discussion was focused on the effect of the ordinary (test) temperature.

Moreover, it is important to note that Wu and Spaepen performed bend tests, not high-precision notch fracture toughness tests, which is likely to potentially induce various additional uncertainties. We showed that (i) When a well-controlled fictive temperature is fixed, the fracture toughness does not exhibit any significant changes with the test temperature, down to very low test temperatures (Fig. 1b) (ii) When the well-controlled fictive temperature is systematically varied for a fixed test temperature, an abrupt toughening transition is observed (Fig. 1a). The first observation seems to disagree with the Wu-Spaepen results, which might suggest that their data correspond to different (and uncontrolled fictive temperatures) or point to a dependence of the thermal DBT on the glass composition. While the second observation might be qualitatively and implicitly consistent with the Wu-Spaepen results, we believe that it is a completely new and fundamental observation, which we also explain theoretically. The theoretical explanation receives additional strong experimental support by demonstrating a directly related transition with the applied strain rate.

In order to clarify that while our observations and their theoretical explanation are completely novel, there were earlier indications in the literature that structural relaxation affects the toughness and ductility of metallic glasses, we add the following sentence (referring once more to Wu and Spaepen) after we report on our main result and highlight its novelty:

“Note, though, that earlier work¹⁸ did indicate that structural relaxation affects the toughness and ductility of MGs”.

The discussion in lines 104 to 109 is confusing. It compares the glass-transition temperature T_g with the fictive temperature T_f , and states that “the difference originates from the pronounced differences in the absolute rates”. However, the value of T_f is set not by the cooling rate, but by the annealing temperature used by the authors. The cooling rate does need to be high enough to preserve the state of the glass, but the cooling rate does not determine T_f . And the difference with the heating rate used to determine the conventional T_g is not really relevant. Heating in the DSC at the conventional rate of 20 K/min is capable of determining values of T_f as well as T_g .

We agree with the reviewer that the text can be improved in this respect. Consequently, we have changed it to read:

“This protocol yields glasses whose metastable structural state is well-characterized by T_f – a temperature characterizing the structural degrees of freedom of the glass where it has fallen out-of-equilibrium, and is often referred to as the fictive temperature^{9,10} or glass-transition-upon-cooling, which is different from the calorimetric glass transition temperature, T_g . T_f is set by the annealing time and is maintained through fast cooling to avoid relaxation to a lower T_f upon cooling. T_g , as typically used for metallic glasses, is determined upon heating with rates of typically 20 K/min (0.3 K/s)”

In lines 132 to 133, the authors claim that a transition in toughness as a function of T_f has not previously been reported. This may strictly be true, but the authors’ finding is not as novel as this statement suggests. The work by Wu & Spaepen (cited above) shows the variation of toughness with annealing, and links the DBTT with the enthalpy of the glass. The enthalpy could easily be linked to T_f .

As discussed in detail above, we do agree that the Wu-Spaepen work has made the first suggestion that structural relaxation may be important for the toughness of glass. This is appreciated and acknowledged in the manuscript. At the same time, as explained above, we strongly believe that our results, both experimentally and conceptually, go well beyond those of Wu and Spaepen.

Since (lines 300 to 301) the authors claim to have developed a procedure “to significantly enhance the fracture toughness of metallic glasses”, it is strange that they do not compare the KQ of their treated samples with the KQ of the freshly thermoplastically formed samples. It is true that the properties of the formed samples may show scatter, but it nevertheless would be good to see a quantitative demonstration of toughening.

We indeed do not compare our toughness measurements, including the toughening associated with the transition we discovered, to the toughness values of as-cast samples. The deep reason for this is given in the general statement appearing in the second paragraph of the manuscript: “A major impediment, however, for their [MGs] widespread usage as a structural material is not their strength, but rather their often low and highly variable fracture toughness”. In particular, the toughness values of as-cast samples corresponding to the MGs employed in our study are typically *lower* than the high-toughness plateau observed in Fig. 1a, but as the reviewer notes the values feature rather large sample-to-sample variability. We do not report these values not only due to their irreproducibility, but mainly because they are likely to be dominated by “extrinsic effects”, e.g. the imprecision and irreproducibility of the notch, stresses, and casting defects, which overshadow the intrinsic toughness¹.

We strongly believe that meaningful comparisons can be made only with highly-reproducible and accurate toughness values, which are obtained in our work and in fact constitute one of the major achievements of the work.

We do agree that the degree of toughness enhancement may vary from glass to glass, and may not always be compelling or “significant”. The more accurate statement is that we provide a well-defined procedure to maximize the toughness for a given glass composition and applied strain-rate by carefully controlling the initial fictive temperature. To reflect this, we have modified the relevant text to read:

“Our results have significant practical implications as they offer a well-defined procedure to realize the maximal well-reproducible fracture toughness of metallic glasses defined by their composition and by the strain-rate in a specific application. Such realization can be achieved by carefully controlling T_f through the annealing protocols described above”.

In lines 160 to 162, the authors note that “surprisingly” KQ has a much stronger dependence on T_f than on the test temperature T. But this is really not surprising. As the authors themselves have noted (lines 99 to 100), there is a variation of more than six orders of magnitude in relaxation times corresponding to the range of T_f . The variation of relaxation time is also reflected in the strong dependence of the liquid viscosity on temperature as the glass transition is approached on cooling. Because the glass (unlike the liquid) is isoconfigurational, the dependence of its viscosity on temperature is much weaker than that of the liquid. Correspondingly, the variation of KQ (or other properties) with test temperature can be rather weak. In other words, the variation of KQ with T_f reflects the temperature dependence of liquid properties, the variation of KQ with T reflects the temperature dependence of glass properties.

We are happy that the reviewer agrees with the physical picture conveyed in the manuscript. However, we stress that “the variation of KQ (or other properties) with test temperature can be rather weak” is at odds with the existing literature (e.g. the Wu-Spaepen paper extensively discussed above) and that the rather abrupt variation of the fracture toughness with a well-controlled fictive temperature and strain-rate is completely new, as well as its theoretical understanding.

In lines 185 to 206, the authors contrast the smooth variation of most properties as a function of T_f with the sharp change seen for KQ. However, the DBT reflects a change in fracture mode and

always appears as a step-like change, even though the underlying properties (e.g. the relaxation time in the glass) vary smoothly.

Even if a change in the fracture mechanism/mode is responsible for our observation, one has to understand *why* this is the case and why it depends on the initial non-equilibrium state of the glass quantified by T_f . While we provide some fractographic evidence for a change in the fracture mechanism across the transition in Fig. 1d, this issue should be further explored in the future (e.g. it is not clear if the change of fracture mechanism is responsible for the transition itself or for the high-toughness plateau). Our work, and the theoretical framework behind it (see Refs. 25-26), highlight the strongly nonlinear nature of plastic deformation near notches and its dynamic nature (leading to the competition of time-scales discussed extensively in the manuscript) as the origin of the transition, without invoking a change in the fracture mechanism. Even if the latter takes place, we strongly believe that it depends on the plastic deformation near the notch root and its effect on the stress state there.

In this work, the authors control the glass structure by annealing (characterized by changes in T_f). In lines 299 to 315, they contrast this type of control with that achieved by varying the cooling rate through the glass transition. In principle, both methods give access to a range of glassy states and there is no fundamental reason why one should correspond to sharp and one to gradual changes in KQ, as would be evident if the same parameter (e.g. T_f) were used to describe the range of glassy states in each case. The authors are certainly correct that the variation of cooling rate may introduce significant scatter, while annealing gives well defined, reproducible states.

We basically agree, though it is not clear that using fixed cooling rates can, even in principle, span the whole range of fictive temperatures achieved in our work. One might need to use more complicated cooling protocols, involving variable cooling rates, with uncertain outcomes.

As already noted, the authors claim that their controlled annealing is a procedure “to significantly enhance the fracture toughness of metallic glasses”. This is potentially misleading. In general, annealing leads towards structural relaxation and embrittlement. So the claimed “toughening transition” (line 154) can in some cases simply be a transition to a state that has undergone less relaxation (in other words a transition towards the as-cast state).

We extensively discussed this point, and the associated change made in the revised manuscript, above. We would like to add here that even if our annealing procedure sometimes bring the glass to “to a state that has undergone less relaxation”, it does *not* by itself imply some rather abrupt transition.

The thermoplastically formed samples may be sufficiently relaxed that the authors’ high- T_f treatments can reduce the degree of relaxation and therefore toughen the samples. As noted above, no experimental evidence has been given for this. Reducing the degree of relaxation by thermal treatments relies on having a high enough cooling rate and has been quantitatively demonstrated by others – Wakeda et al., Controlled rejuvenation of amorphous metals with thermal processing, Scientific Reports 5 (2015) 10545.

We already addressed above the uncertainties associated with the as-cast samples. Moreover, even if the scenario the reviewer refers to is relevant to our experiments, i.e. that our annealing procedure reduces the degree of relaxation of the glass, this by itself does not explain why the toughening transition is rather abrupt and not gradual.

In Fig. 4 and the associated discussion, the authors attempt to demonstrate a degree of correspondence between the conventional glass transition and the DBT, which is likened to a mechanical glass transition. This may be striving too hard to demonstrate some new fundamental insight that is not there. In particular, the attempts to make Figs 4a, 4b and 4c look similar to each other distorts observed behavior and is misleading.

In the conventional glass transition (Fig. 4a), the property of the glass lies ABOVE the extrapolated liquid line at the same temperature – i.e. in terms of the liquid line, the glass has properties at a given temperature that reflect an effective temperature that is higher than the actual temperature. For the claimed mechanical glass transition, however (Figs 4b and 4c), the reverse is true – which clearly reflects different behavior.

In Fig. 4a the enthalpy line for the glass is shown curved, presumably to make an analogy with the toughness lines for the glass in Figs 4b and 4c. But, for actual glasses, the enthalpy line is not noticeably curved. The key problem is that the DBT depicted as toughness versus temperature show step-like behavior, quite distinct from the ‘change of slope’ behavior of the glass transition. If, in Fig. 4a, the authors were to plot specific heat capacity rather than enthalpy, then a better analogy (at least superficially) would emerge.

Our goal in raising the analogy between the toughening transition we discovered and the conventional glass transition, as explain in the manuscript, is to highlight to things:

- (1) The transition temperatures appear to be close, $\frac{T_f^{DB}}{T_g} \sim 1$.
- (2) More importantly, the two phenomena appear to emerge from a competition between an externally applied rate and an intrinsic relaxation rate.

While we are not sure the analogy is deep, we feel that it might be illuminating and stimulating to consider it.

Finally, we insist that our results, both experimental and conceptual, are deep, fundamental and highly novel. We believe they will be of interest to a broad range of scientists and will trigger a lot of additional research.

In the caption to Fig. 1a, there is mention of an inset, but there is no inset in the figure.

We removed the sentence about the inset, which was a leftover from the original manuscript and figures. Many thanks.

We thank the reviewer again for her/his time, effort and detailed comments that helped us improving the manuscript.

Reviewer #3:

The - entirely - revised version of the paper proposed by Ketkaew et al., titled Mechanical glass transition revealed by the fracture toughness of metallic glasses, provides new great insights of fundamental as well as practical importance.

All the commentary I had made for the first version have been answered, and beyond.

We thank the reviewer for these encouraging comments.

I see only a few points to make:

1) Line 160-162, the authors state that the variations of K_Q with T_f are 100 times larger than the variation of K_Q with T . I don't see how that follows from their data. They write : "100 times larger than the negligible variation of K_Q with T over the same temperature range" : this expression should be clearly justified. What range are the authors considering and what are their reasons to consider this range particularly ? Depending on the considered range, fig 1.b might show no sign of a 100 ratio between the variation of K_Q with T with respect to its variation with T_f . Indeed, for high T_f materials, the measured variation of K_Q is approximately $110-60 = 50$, whereas the DB variation with T_f is approximately $110-40 = 70$, which is at best a factor 2 ratio. It's just that the BDT controlled by T or T_f occur at different temperatures and the control of ductility by one parameter or another (T or T_f) should be made between the highest and lowest value of K_Q .

We agree. The main point we wanted to convey is that the toughness changes much more dramatically with the fictive temperature over the same temperature range (where indeed there is a factor of ~ 100 between the two) in order to highlight the structural (as opposed to thermal) origin of the transition we observed.

To clarify the point, we changed the text to read:

“We find that the variation of K_Q with T_f is significantly larger than the negligible variation of K_Q with T over the same temperature range (Fig. 1b), highlighting the structural nature of the transition.”

2) In spite of the great care brought to experimental measurements, the rather small number of data points in the cryogenic part of fig 1.b does not permit to know whether the whole thermal ductile-brittle transition has been captured. If I had to take a bet, my guess would be that no and that measurements at even lower temperatures might reveal that the high T_f curve (circles) converges with the low T_f (triangles) in fig 1.b. But that is just a guess and maybe the authors have additional data or information on that particular point. I don't think additional experiments

are absolutely necessary to verify this, though, as it is not the center of the present study, but I think it should be included in the discussion to mitigate the conclusions. As a matter of fact, it is impossible to say, from these data (3 points), if the whole thermal BDT has been measured and conclusions cannot be made too positively on that particular point.

The goal of measuring the toughness also as a function of the ordinary (test) temperature was to rule out the possibility that our main observation is related to any thermal BDT. We had no intention to study in detail the thermal BDT that appears to occur at much lower temperatures (we had to go down to the liquid nitrogen temperature in order to observe a significant variation in the toughness).

To clarify this point further, we added the red text to the part in which we compare the sensitivity to both the fictive and ordinary temperatures:

“These results lead us to conclude that the fracture toughness of MGs is qualitatively and dramatically more sensitive to the non-equilibrium structural state of the glass quantified by T_f than to the (ordinary) temperature T (at least down to very low ordinary temperatures, where another transition might take place, see the red data points at the liquid nitrogen temperature in Fig. 1b)”.

We also removed “... but not as a function of T ” from the caption heading of Fig. 1 as we cannot explain in the heading which temperature range we are referring to.

3) In a very rough manner, I think it would more appropriate to consider that T_f seems to control an athermal or structural value of K_q in the ductile regime (so to speak) and that there is still a thermal ductile brittle transition, as the data presented by the authors seem to suggest. This point of view might have also implication on the 100 ratio discussion in point 1. Moreover, it seems to me that T_{BDT} appears independent of T_f , still from fig 1.b, which could be made explicit (in a short manner, perhaps) if this seems correct to the authors.

We absolutely agree that the fictive temperature controls the structure of the glass, which affects its ductility and toughness. This does not rule out the existence of a thermal BDT transition at very different temperatures, as indeed our data appear to suggest. The revision discussed in the previous point addresses this issue.

The properties of the thermal BDT transition, for example its sharpness and dependence on the fictive temperature, cannot be reliably extracted from the available data (i.e. it not sufficiently densely sampled as a function of T , as the reviewer notes). Therefore, while we agree that this point is interesting (e.g. the Wu-Spaepen work discussed above suggests some dependence of the thermal BDT transition on structural relaxation), we prefer not to make any explicit statements in this context in the manuscript (though the data are there and can be judged).

Beyond these 3 points, I think this paper of really great value, I am unaware of any similar results having ever being published (or I'd like to be clearly pointed to them if they exist) and I highly recommend its publication in Nature Comm.

We are grateful to the reviewer for identifying the merit of our work and for the recommendation to publish it in Nature Communications.

To conclude, we have thoroughly addressed all of the reviewers' comments and suggestions, and made the corresponding changes in the revised manuscript. We hope that the revised manuscript is now suitable for publication in Nature Communications.

1. Chen W, *et al.* Processing effects on fracture toughness of metallic glasses. *Scripta Mater* **130**, 152-156 (2017).

REVIEWERS' COMMENTS:

Reviewer #1 (Remarks to the Author):

Most of my suggestions have been addressed. However, some minor revisions still need to be made before formal acceptance, e.g.,

The statement in the abstract "... T_f , which characterizes glass structure." is confusing.

Actually T_f is a measure of the average level of a liquid on the potential energy landscape.

The explanation about the difference between T_f and T_g is not fully correct.

It is good that the authors added a figure illustrating their T_f determination.

Reviewer #2 (Remarks to the Author):

The authors have made extensive changes to take account of the reviewers' comments. The key issues have been dealt with, and in some cases this has just meant softening the original claims.

The reviewer is still not convinced that the suggested analogy between the toughening transition and the conventional glass transition is useful -- but the authors can be allowed to make their case in print.

The revised manuscript is now acceptable for publication without further changes.

Reviewer #3 (Remarks to the Author):

The authors have, in my opinion, satisfactorily answered all questions and I recommend publication under this revised form.